# CHILDREN'S INTELLIGENCE TESTS POSE CHALLENGES FOR MLLMS? KIDGYM: A 2D GRID-BASED REASONING BENCHMARK FOR MLLMS

**Hengwei Ye[1], Yuanting Guan[1], Yuxuan Ge[1], Tianying Zhu[1], Yijia Zhong[1], Yijing Zhang[1], Han Zhang[1], Yingna Wu[1], Zheng Tian[1]***

[1]ShanghaiTech University

## ABSTRACT

Multimodal Large Language Models (MLLMs) combine the linguistic strengths of LLMs with the ability to process multimodal data, enabling them to address a broader range of visual tasks. Because MLLMs aim at more general, human-like competence than language-only models, we take inspiration from the Wechsler Intelligence Scales — an established battery for evaluating children by decomposing intelligence into interpretable, testable abilities. We introduce **KIDGYM**, a comprehensive 2D grid-based benchmark for assessing five essential capabilities of MLLMs: **Execution, Perception Reasoning, Learning, Memory, and Planning**. The benchmark comprises 12 unique tasks, each targeting at least one core capability, specifically designed to gauge MLLMs' adaptability and developmental potential, mirroring the stages of children's cognitive growth. Additionally, our tasks encompass diverse scenarios and objects with randomly generated layouts, ensuring a more accurate and robust evaluation of MLLM capabilities. **KIDGYM** is designed to be fully user-customizable and extensible, allowing researchers to create new evaluation scenarios and adjust difficulty levels to accommodate the rapidly growing MLLM community. Through the evaluation of state-of-the-art MLLMs using **KIDGYM**, we identified significant insights into model capabilities and revealed several limitations of current models. We release our benchmark at: `https://kidgym.github.io/KidGym-Website/`.

## 1 INTRODUCTION

Large language models (LLMs) have demonstrated significant success across language-based tasks (Brown et al., 2020; Sharan et al., 2023), laying a strong foundation for advancements in artificial intelligence. Following this success, multimodal large language models (MLLMs), which integrate multiple data modalities such as images (Wang et al., 2024f) and videos (Cai et al., 2024; Wang et al., 2024c), are also experiencing rapid growth and development. By fusing diverse information sources, MLLMs enable AI to learn and reason (Gao et al., 2024) across different modalities, bringing it closer to human-like cognition (Du et al., 2024).

Human cognitive testing frameworks (Smith & Gasser, 2005) have been well-established and refined over time, and insights from cognitive developmental psychology have consistently provided valuable guidance in advancing artificial intelligence (Lake et al., 2016; Wu et al., 2024a; Sumers et al., 2024; Salas-Guerra, 2025). Research in psychometric AI and universal psychometrics shows that ability-oriented batteries adapted from validated human tests offer a principled way to gauge general reasoning, outperforming task-specific benchmarks (Voudouris et al., 2024; 2025). Within this context, the shift from LLMs to MLLMs calls for an evaluation framework that profiles multiple coordinated abilities rather than language alone (Wang et al., 2024a; Han et al., 2025; Gopnik et al., 2009), aligning well with child intelligence assessments such as the Wechsler scales, suggests that evaluating MLLMs with child-focused cognitive frameworks is a particularly promising approach.

While Wechsler framed human intelligence as a constellation of interrelated abilities that vary in degree, the cognitive profile of MLLMs cannot be mapped one-to-one onto these human con-

---

*Correspondence to Zheng Tian <tianzheng@shanghaitech.edu.cn>

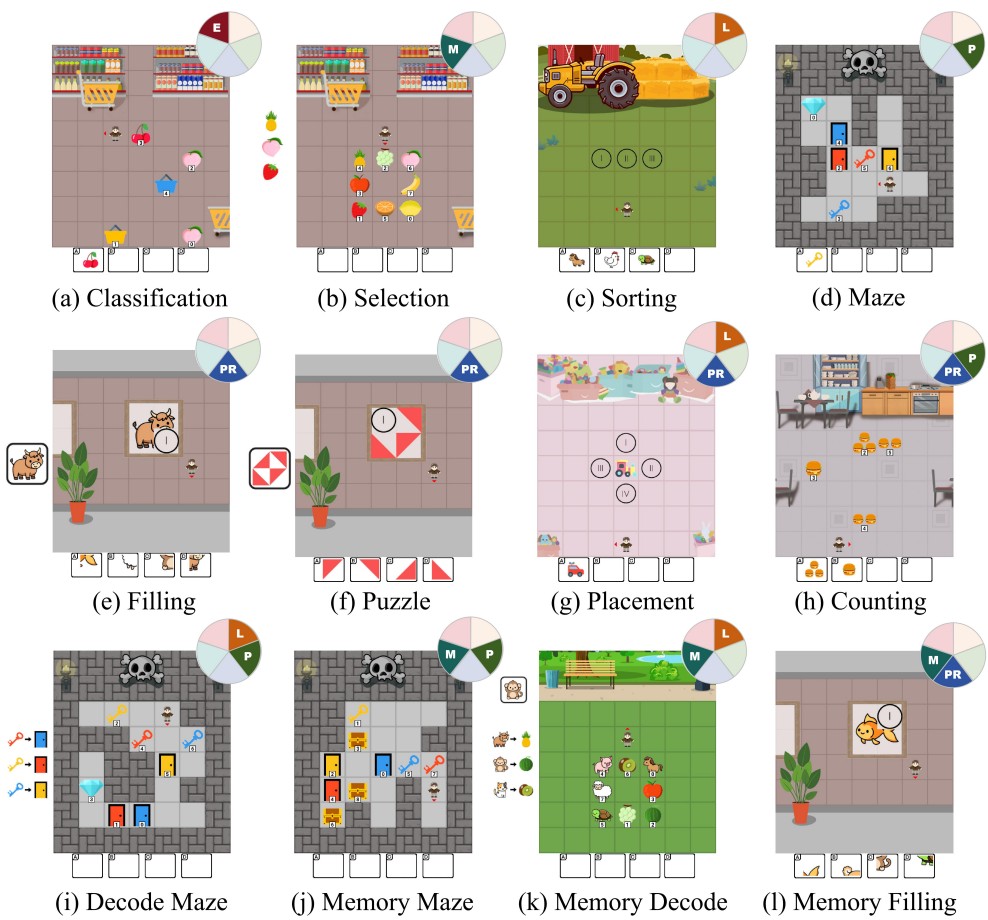

(a) Classification    (b) Selection    (c) Sorting    (d) Maze

(e) Filling    (f) Puzzle    (g) Placement    (h) Counting

(i) Decode Maze    (j) Memory Maze    (k) Memory Decode    (l) Memory Filling

Figure 1: Previews of 12 tasks in **KIDGYM**. The circular chart in the upper-right corner of each subfigure represents the cognitive abilities required by the task: **E** for **Execution**, **M** for **Memory**, **L** for **Learning**, **P** for **Planning**, and **PR** for **Perception Reasoning**.

structs (Bender & Koller, 2020). Consequently, each MLLM capability requires a bespoke definition that respects the model's architectural and functional particularities (Binz & Schulz, 2023).

In this work, we introduce **KIDGYM**, a new benchmark specifically designed for evaluating MLLMs' cognitive abilities. Drawing inspiration from the Wechsler Intelligence Scales (Guertin et al., 1966; Zhu et al., 2004; Zeigler-Hill et al., 2020), a widely recognized children's intelligence test, we summarized and defined five essential capabilities that MLLMs require in the current state: **Execution, Perception Reasoning, Memory, Learning, and Planning**.

**KIDGYM** comprises 12 carefully designed tasks: 6 focused on testing individual capabilities and 6 on assessing integrated dual capabilities. To ensure robust and reliable experimental results, our tasks cover a wide range of scenarios and objects with randomly generated layouts. Furthermore, to evaluate the performance limits of various MLLMs, each task is presented at three difficulty levels (L1, L2, L3) from easy to hard. In order to support customization, we built the benchmark based on the Gym API (Brockman et al., 2016), allowing researchers to create new evaluation scenarios to accommodate the rapidly growing MLLM community.

We benchmark a representative set of state-of-the-art MLLMs on **KIDGYM**, including closed-source models: o3 (OpenAI, 2025a), GPT-5 (OpenAI, 2025b), GPT-4o (OpenAI, 2024), Gemini-2.5-Pro (DeedMind, 2025), Gemini-2.5-Flash (DeepMind, 2025), Claude-3.7-Sonnet (Anthropic, 2025) and strong open-source models: DeepseekVL-2 (Wu et al., 2024c), QwenVL-2.5 (Bai et al., 2025), InternVL-3 (AILab, 2025) with different sizes.

Through systematic experiments, closed-source models can achieve near-perfect scores on specific tasks and excel particularly in learning tasks. Among these, o3, GPT-5 and Gemini-2.5-Pro significantly outperforms the other models by a significant margin across all capability dimensions.

However, we identified 4 key challenges for current MLLMs not captured in previous benchmarks:

- First, models show **limitations in reasoning over non-semantic, abstract visual information**.
- Second, models are **not sensitive to item quantity**.
- Third, models **struggle with composite capacity tasks** involving the interaction of multiple rules.
- Finally, models perform relatively **poorly on perception reasoning and planning tasks**.

Our contributions can be summarized as follows:

1) We propose an assessment framework for MLLMs, incorporating five core abilities based on the Wechsler Intelligence Scale.
2) We introduce **KIDGYM**, a unified 2D benchmark for MLLMs, featuring diverse environments, randomized layouts, graded difficulty levels, and customization options.
3) We conduct a systematic evaluation of state-of-the-art MLLMs, highlighting empirical strengths and weaknesses, and providing insights for future development.

## 2 RELATED WORKS

### 2.1 MULTIMODAL LARGE LANGUAGE MODELS

LLMs (Ouyang et al., 2022; Touvron et al., 2023; Chung et al., 2024) have evolved from processing solely text-based inputs to exhibiting multimodal capabilities. This advancement has significantly expanded the applicability of MLLMs in areas such as image description (Liu et al., 2016; Tan et al., 2024), image reasoning (Ilievski & Feng, 2017; Wang et al., 2024e; Xiao et al., 2024), and visual question answering (VQA) (Gaur et al., 2024; Wang et al., 2024b), bringing us closer to the ultimate goal of AI research: general artificial intelligence (AGI) (Zhong et al., 2024), which aims to develop systems capable of matching or surpassing human-level performance across diverse domains.

### 2.2 MLLM BENCHMARK

Numerous benchmarks have been developed over time to assess the capabilities and performance of MLLMs. Initially, these benchmarks primarily focused on evaluating MLLMs' ability to process and understand multi-modal data, such as image comprehension and analysis (Li et al., 2023; Xu et al., 2023; Yin et al., 2023; Yu et al., 2023; Fu et al., 2024). As MLLMs demonstrated proficiency in recognition tasks (Kuchibhotla et al., 2024), attention shifted toward evaluating their reasoning abilities (Shi et al., 2024; Han et al., 2023), including inductive, deductive, and abductive reasoning (Huang & Zhang, 2024). Advancing further, a diverse range of specialized benchmarks has emerged, focusing on various aspects of MLLM capabilities across different application scenarios (Li et al., 2024), including creativity in image generation (Fang et al., 2025), information processing in long contexts (Song et al., 2024) and human-level planning in real-world problems (Chen et al., 2024), highlighting the rapid evolution of MLLM field.

However, current MLLM benchmarks predominantly evaluate static tasks — where information remains constant throughout (Amini-Naieni et al., 2024; Cao et al., 2024), rather than dynamic tasks requiring continuous environmental interaction and adaptation (Xu et al., 2024). Dynamic tasks require the agent to follow a trajectory or execute a sequence of actions through continuous interaction to ultimately complete the task, rather than answering in the simple question-and-answer format typical of static tasks (Gonzalez, 2005). Furthermore, most existing benchmarks typically assess isolated capabilities (Krishna et al., 2025), providing limited insight into how the diverse competencies of MLLM compare or interact in real-world contexts (Tihanyi et al., 2024). Positioning **KIDGYM** against this backdrop, Table 1 provides a structured comparison with representative benchmarks.

One promising approach to addressing these limitations is the use of games as benchmarking tools (Juliani et al., 2019; Samvelyan et al., 2021; Gan et al., 2021). Games offer dynamic, multi-

Table 1: Comparison of **KIDGYM** with existing benchmarks across target paradigm, difficulty-level support, user extensibility, evaluated capabilities, and dynamic vs. static settings.

| Benchmarks | Target | Difficulty Level | User Extensible | Capabilities | Dynamic/Static |
|---|---|---|---|---|---|
| Crafter | RL | ✗ | ✗ | / | Dynamic |
| MiniGrid | RL | ✓ | ✓ | / | Dynamic |
| LogicGame | LLM | ✓ | ✗ | Learning Planning Execution | Static |
| EgoPlan | MLLM | ✗ | ✗ | Planning | Dynamic |
| MileBench | MLLM | ✗ | ✗ | Memory | Static |
| Countgd | MLLM | ✗ | ✗ | Counting | Static |
| CompBench | MLLM | ✗ | ✗ | Reasoning | Static |
| MaRs-VQA | MLLM | ✗ | ✗ | Reasoning | Static |
| ARC-AGI-2 | MLLM | ✗ | ✗ | Reasoning(Abstract) | Static |
| **KIDGYM** | MLLM | ✓ | ✓ | All Above | Dynamic |

dimensional environments that can better simulate complex, interactive tasks. For instance, Mini-Grid (Chevalier-Boisvert et al., 2023) provides a suite of goal-oriented game environments, but it was originally designed for reinforcement learning. SmartPlay (Wu et al., 2024b) incorporates six classic games, including Minecraft (Johnson et al., 2016) and Crafter (Hafner, 2021), converting gameplay scenarios into text-based descriptions to evaluate key LLM capabilities such as instruction following and error correction. While tabletop games (Costarelli et al., 2024), logical games (Gui et al., 2024) and board games (Topsakal et al., 2024) have been utilized for model evaluation, these approaches predominantly focus on text-based assessments and are less suitable for evaluating MLLMs.

To fill these gaps, we introduce **KIDGYM**, a suite of interactive and dynamic scenes, which not only assess individual capabilities but also enable the simultaneous evaluation of multiple dimensions of intelligence (e.g., memory and planning). This integrated evaluation offers a more accurate and comprehensive understanding of MLLMs' strengths and weaknesses.

## 3  CAPABILITIES

Humans and MLLMs differ fundamentally in their embodiment and interaction modalities, so a literal transfer of abilities and subtests is inappropriate. Given these limitations, we do not simply copy the Wechsler test. Throughout the development of **KIDGYM**, we collaborated with co-authors who are experts in child brain science and translated the most important Wechsler indicators into five core competencies that are critical for MLLMs.

**Execution:** Children's behavior is widely viewed as the realization of prior intentions (Searle, 1983). In cognitive science, this capacity is captured by executive function (Diamond, 2013) — the conscious regulation of thought and action. Analogously, MLLMs must translate internal representations of goals into concrete behaviors to produce meaningful outcomes. We therefore define execution as the capability of an MLLM to fulfill a task on the basis of its inferred goals and constraints. Whether the model is navigating a virtual world, manipulating physical objects, or coordinating with other agents, robust execution bridges the gap between abstract objectives and verifiable behavior, ensuring that understanding is consistently transformed into successful action.

**Memory:** Memory allows the human to encode, store, and retrieve information so that past experiences guide current decisions (Atkinson & Shiffrin, 1968). MLLMs, however, can reread the entire interaction history at every step; their "memory" therefore emphasizes maintaining long-range contextual dependencies rather than reconstructing fragmented episodes. Throughout this paper we define an MLLM's memory as its capacity to retain previously perceived information, integrate that information into a coherent context, and exploit the evolving context to refine subsequent reasoning

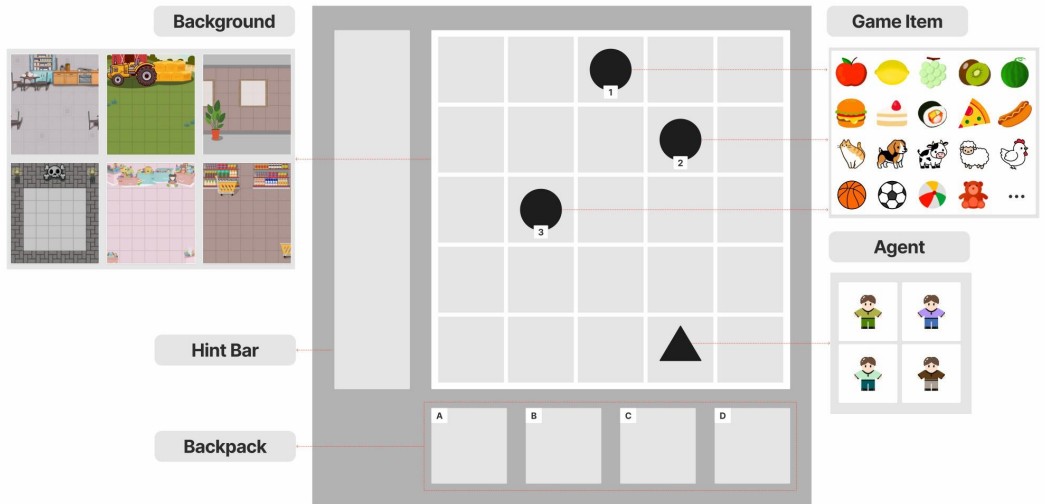

Figure 2: A **KIDGYM** task frame comprises a scene map, a backpack, and a hint bar. We provide varied agent skins, backgrounds, and scene-specific items; backpack slots and in-scene items are letter/number-labeled for identification. Resolution and grid layout are specified in Appendix B.1.

and actions (Wang et al., 2024d). Such persistent memory is indispensable for tasks that demand sequential understanding and consistent decision-making across multiple turns (Zhang et al., 2024).

**Learning:** Learning is the process by which an individual acquires new knowledge, skills, attitudes or behaviors through experience, practice or formal education. The capacity to learn is a central cognitive ability that distinguishes humans from most other species. In MLLMs, learning refers to the model's capacity to ingest previously unseen information or rules and effectively apply them in decision-making and problem-solving (Huo & Tang, 2025; Tai et al., 2024). A major challenge emerges when the incoming information conflicts with, or supersedes, what the model has already stored. Without further fine-tuning, an MLLM must reconcile such inconsistencies on-the-fly. Real-world tasks are dynamic: new constraints, updated facts and evolving user goals continually arise. Therefore, an MLLM that can learn, adapt and deploy fresh knowledge without frequent and costly retraining will be more flexible, robust and economically viable in practice.

**Planning:** In human intelligence, planning serves as a fundamental cognitive process that enables individuals to anticipate outcomes, formulate strategies, and sequence actions to achieve desired objectives. Within the context of MLLMs, planning constitutes the capacity to systematically organize tasks, predict action consequences, and implement multi-step strategies for complex problem-solving (Zheng et al., 2024). This capability transcends mere reactive decision-making by incorporating foresight—requiring models to balance immediate actions against long-term goals while navigating the inherent trade-offs between short-term responses and strategic outcomes.

**Perception Reasoning:** In the Wechsler, perceptual reasoning measures the ability of children to solve purely visual problems, integrating spatial perception, visual organization, and nonverbal reasoning. By analogy, we define perception reasoning in the context of MLLMs as the capability to draw inferences and make decisions directly from visual inputs (Xiao et al., 2025). This capability goes beyond object recognition: the model must analyze visual evidence, construct a coherent chain of logic, anticipate plausible outcomes, and choose actions that follow from those predictions.

## 4 MECHANICS

The tasks in **KIDGYM** have been specifically designed with several mechanisms (see Figure 2) that take into account both the strengths and weaknesses of current MLLMs.

**Diverse Semantic Scenes:** In real-world applications, tasks of the same type often vary based on their contextual scenarios. To capture these variations, we have designed a range of environments, including supermarkets, canteens, and farms, along with corresponding items to create immersive,

context-rich scenarios. By evaluating the model in our original contexts, where the context is randomized for most task types, we can assess whether it has acquired the targeted abilities and can apply them effectively across varying scenarios, rather than relying on memorization of similar environments from pretraining data. This helps mitigate data leakage or contamination to some extent.

**Randomness:** In addition to diverse semantic scenes, variability in task layouts is crucial for assessing MLLM robustness. While semantic diversity introduces novel contexts across tasks, layout stochasticity generates distinct configurations within the same task and scene. Each episode initializes with randomized element arrangements (e.g., item locations, agent spawn), ensuring no two rounds are identical. This randomness reduces evaluation variance and ensures more consistent performance estimates. The exact question counts are provided in Appendix B.2.

**Backpack and Hint Bar:** Current MLLMs often struggle to maintain contextual consistency (Chou et al., 2024), particularly when dealing with hidden details not explicitly represented in visual information. For example, an agent may successfully '*pick up the key*' in one step but fail to recall possessing it in later steps. To address this problem, we designed a backpack and a hint bar as components of the task state, enabling agents to retrieve crucial information through out the task.

**High-level Actions:** MLLMs are not well-suited for executing atomic actions such as "*go one step forward*" or "*turn left*" in tasks requiring high operability. In contrast, MLLMs are better suited for handling macroscopic concepts and executing high-level actions. Building on this, each task in **KIDGYM** presents MLLMs with high-level actions. For instance, the agent can directly perform actions such as "*pick up the basketball*" instead of navigating step-by-step to its location and interacting with it. This reduction in operational granularity enables the model to focus on actions that are directly tied to meaningful outcomes, avoiding low-level controls.

**Identification:** Each item in **KIDGYM**'s task scenes is assigned unique identifiers. These identifiers enable the MLLMs to associate visual elements with text-based descriptions in high-level actions or goals, such as "*put the item from backpack A into item number 2.*" These labels not only optimize information retrieval but also reduce ambiguity in task execution, ensuring that the agent interprets and interacts with the environment accurately.

## 5 TASKS

We design 12 tasks to evaluate MLLMs: 6 targeting a single capability and 6 targeting composite capabilities. Each task includes three difficulty levels, from easy to hard. **KIDGYM** follows standard psychometric practice like Wechsler by constructing tasks in which one (or two) target abilities are dominant by design. Detailed information for each task can be found in Appendix F.

### 5.1 SINGLE CAPACITY TASK

**Classification (CL):** In CL task, the agent is required to place each item into its designated container based on specific instructions, such as "*placing the cherry in the yellow basket*" (see (a) in Figure 1). It is designed to evaluate the MLLM's **Execution** ability, which involves translating an understanding of goals into effective actions. The agent's performance in this task measures its accuracy in following instructions within a structured environment.

**Selection (SE):** In SE task, several random items will appear in the left hint bar at first (see (b) in Figure 1). Once the task starts, these items will be hidden, and the agent need to select the items that appeared in the hint bar before. This task evaluates the MLLM's **Memory** capability by requiring it to remember and recall the items previously shown.

**Sorting (SO):** In SO task, the agent is presented with a rule that may contradict real-world knowledge. For instance, the agent might be instructed that "the faster the animal, the heavier it is". The agent is expected to correctly rank the animals based on the given rule (see (c) in Figure 1). This task evaluates the MLLM's **Learning** abilities, as it requires the agent to comprehend a novel rule that may conflict with its prior knowledge.

**Maze (MA):** This task is inspired by Procgon (Cobbe et al., 2019), where the agent must obtain the diamond in a maze with several locked doors. The agent needs to collect the corresponding colored

keys to unlock these doors (see (d) in Figure 1). This task primarily evaluates the MLLM's **Planning** ability, as the agent should carefully devise a strategy to reach the diamond with the fewest steps.

**Filling (FI):** In FI task, the agent will be presented with an image in which a quarter section has been removed, such as "*a goldfish with a missing head*" (see (e) in Figure 1). Then it needs to restore the image by selecting the correct missing piece from a set of distractors in the backpack. This task primarily evaluates the MLLM's **Perception Reasoning** ability, as it requires the agent to develop a holistic understanding of the image and infer the missing part.

**Puzzle (PU):** In PU task, a target image composed of 4 puzzle pieces is displayed in the hint and the agent needs to assemble the scattered puzzle pieces from its backpack to reconstruct the target (see (f) in Figure 1). This task primarily evaluates the MLLM's **Perception Reasoning** in abstract visual mode, as it requires the agent to grasp the image's overall structure, which cannot be easily conveyed through language.

## 5.2 COMPOSITE CAPACITY TASK

**Placement (PL):** In PL task, the agent is required to place the item in the opposite position based on the given goal. For instance, if the rule states "place the toy car on the north side of the toy train" (see (g) in Figure 1), the agent actually needs to place it on the "south" side. This task primarily evaluates the MLLM's abilities in **Learning** and **Perception Reasoning**, as it necessitates an understanding of placement rules and the awareness of spatial orientation.

**Counting (CO):** In CO task, the scene contains several piles of items, with quantities ranging from 1 to 3 (see (h) in Figure 1). At the start of the task, the agent is given a target number and then it must collect exactly that number of items. This task primarily evaluates the MLLM's **Perception Reasoning** and **Planning** abilities, focusing on the agent's awareness of item quantities and its strategic decision-making regarding how many items to collect at single time.

**Decode Maze (DMA):** This task follows the same rules as the "**Maze**", with an added challenge. The agent can no longer use a same-colored key to open a door. Instead, it must learn the "key–door" correspondence shown in the hint bar (see (i) in Figure 1), such as "use the blue key to open the yellow door". This task evaluates the MLLM's **Learning** and **Planning** abilities, requiring the agent to leverage the hint information to make correct choices and formulate a series of plans to obtain the diamond as few steps as possible.

**Memory Maze (MMA):** This task follows the same rules as the "**Maze**", with an added challenge. Before the task begins, the agent is shown the location of the diamond, but once the task starts, the diamond in the scene will be hidden and several treasure chests will appear (see (j) in Figure 1). To succeed, the agent must correctly open the chest containing the diamond. This task primarily assesses the MLLM's **Memory** and **Planning** abilities, as the agent must recall the diamond's location and devise an effective strategy to retrieve it.

**Memory Filling (MFI):** This task follows the same rules as "**Filling**", with an added challenge. The agent must additionally remember the target, which will disappear once the task starts (see (k) in Figure 1). This task primarily evaluates the MLLM's abilities in **Perception Reasoning** and **Memory**, as it necessitates recognizing the overall image and recalling specific details to identify the correct piece.

**Memory Decode (MDE):** In MDE task, the agent is provided with a hint bar, which contains a certain number of association rules between different items (see (l) in Figure 1) and it must remember the item relationships because these will be hidden once the task starts. This task evaluates the MLLM's abilities in **Memory** and **Learning**, as it requires the agent to retain and utilize the information from the hint bar to make accurate selections.

## 6 EXPERIMENTS

### 6.1 EXPERIMENTAL SETUP

We evaluated 9 state-of-the-art MLLMs on **KIDGYM**, covering both closed-source and open-source models. The closed-source models are: o3 (OpenAI, 2025a), GPT-5 (OpenAI, 2025b),

GPT-4o (OpenAI, 2024), Gemini-2.5-Pro (DeedMind, 2025), Gemini-2.5-Flash (DeepMind, 2025), Claude-3.7-Sonnet (Anthropic, 2025), while the open-source models are DeepSeekVL-2 (Team, 2024), QwenVL-2.5 (Bai et al., 2025), and InternVL-3 (AILab, 2025). For comparison, we also provide human and random baselines (see Appendix C).

## 6.2 EXPERIMENTAL METRICS

We evaluate closed-source models via their official APIs and open-source models using NVIDIA RTX A6000 GPUs. For each task, we ran 100 zero-shot rounds and evaluated every model on the identical set (see Appendix B.6 for detailed evaluation procedure). We also tested chain-of-thought (CoT) and in-context learning (ICL) methods on part of the tasks and models, with results presented in Appendix E.

## 6.3 EXPERIMENTAL RESULTS

In this section, we compare the performance of MLLMs on **KIDGYM**, as presented in Table 2. The performance here is measured by the success rate under the ground-truth optimal solution.

From the results, the overall performance of closed-source MLLMs is significantly higher than that of open-source ones and most models perform better in tasks that examined a single ability than in tasks that involved composite abilities. Notably, three of the currently most powerful closed-source models (o3, GPT5 and Gemini-2.5-Pro) are able to achieve near-perfect scores on a few specific tasks, such as CL, SE and MDE. However, from a difficulty perspective, success rates generally decrease from L1 to L3, validating the effectiveness of our task taxonomy. Through quantitative analysis, we identified 3 main challenges of current MLLMs.

**Challenges in Reasoning over Non-Semantic Visual Information.** Both FI and PU tasks require the model to reassemble pieces to match a target image. In FI task, the target image contains recognizable and nameable objects (e.g., animals). In PU task, however, the model must reconstruct an arbitrary shape made of random blocks. The highest success rate for the FI-L1 task is 0.83 (o3), while the highest success rate for the PU-L1 task is only 0.30 (GPT-5), which is merely 5 percentage points higher than the random success rate. Across all models, performance of PU task is consistently worse than of FI task, suggesting that frontier MLLMs still struggle with abstract, non-semantic images.

**Challenges in Identifying the Quantity of Items.** The goal of the CO task is to collect a specific number of items in the scene, which is very easy for humans (the human success rates for all three difficulty levels is 1.00). However, the CO task appears to present significant failure for current MLLMs, even the best model, Gemini-2.5-Pro, achieves only a 0.72 success rate on the easiest level (L1). In most failure cases, the model conflates a small cluster of items (typically two or three) and identifies them as a single object. Furthermore, we conducted a small-scale experiment and found that increasing image resolution improved several models' accuracy on the CO task. In contrast, humans can complete the task without enhanced image clarity. This suggests that current MLLMs are insufficiently sensitive to quantitative information and tend to rely on high-resolution visual cues rather than robust numerosity representations. Detailed results are provided in the Appendix B.1.

**Challenges in Dealing with Composite Tasks.** Compared with tasks that call for a single capability, success rate drops markedly on several tasks that demand a combination of abilities. For example, MMA/MFI tasks extend the original MA/FI tasks by adding a memory requirement. The success rates of all the models in MMA/MFI tasks are significantly lower than that of the MA/FI tasks, which assesses a single capability. Therefore, for MLLMs, it remains a challenge to process multiple types of information at once or to take into account interrelated rules simultaneously.

**Reasoning Method has Significant Impact on Different Tasks.** By comparing 3 methods (zero-shot, CoT, ICL), we observe significant differences in their success rates across various tasks. For instance, when using CoT, the Gemini-2.5-Flash model demonstrates remarkable improvements over zero-shot. However, for o3, which inherently integrates the CoT, no substantial difference is observed between zero-shot and CoT. In the case of ICL, where scene and item layouts are randomly generated, its performance may even be inferior to zero-shot in certain tasks that emphasize memory and learning. This could be attributed to the model's tendency to overemphasize examples, potentially neglecting the dynamic changes within the scene.

Table 2: Zero-shot performance comparison of MLLMs across 12 **KIDGYM** tasks. **"L"** denotes the task level. Performance is measured by the success rate over 100 rounds under the **ground-truth optimal solution**, rounded to two decimal places.

| Methods | L | CL | SE | SO | MA | FI | PU | PL | CO | DMA | MMA | MDE | MFI |
|---|---|---|---|---|---|---|---|---|---|---|---|---|---|
| *Closed-Source Models (API)* | | | | | | | | | | | | | |
| o3 | 1 | **1.00** | **1.00** | 0.97 | 0.87 | **0.83** | 0.26 | **1.00** | 0.30 | 0.90 | 0.44 | **1.00** | **0.81** |
| | 2 | 0.98 | **1.00** | 0.95 | 0.42 | 0.52 | 0.11 | 0.98 | 0.26 | **0.47** | 0.18 | **1.00** | 0.50 |
| | 3 | 0.92 | **1.00** | **0.97** | **0.27** | 0.30 | 0.06 | 0.71 | 0.13 | 0.17 | 0.05 | **1.00** | 0.37 |
| GPT-5 | 1 | **1.00** | **1.00** | **1.00** | **0.97** | 0.74 | **0.30** | **1.00** | 0.36 | **0.95** | 0.62 | **1.00** | 0.77 |
| | 2 | 0.96 | **1.00** | **0.99** | **0.43** | 0.60 | 0.06 | **1.00** | 0.18 | **0.47** | 0.10 | **1.00** | 0.61 |
| | 3 | 0.92 | 0.99 | 0.94 | 0.11 | **0.41** | 0.01 | **0.88** | 0.16 | **0.24** | 0.01 | **1.00** | **0.40** |
| GPT-4o | 1 | 0.46 | **1.00** | 0.48 | 0.33 | 0.66 | 0.26 | 0.71 | 0.00 | 0.58 | 0.00 | 0.95 | 0.64 |
| | 2 | 0.24 | 0.76 | 0.31 | 0.19 | 0.28 | 0.09 | 0.37 | 0.00 | 0.14 | 0.00 | 0.98 | 0.26 |
| | 3 | 0.13 | 0.50 | 0.08 | 0.00 | 0.15 | 0.03 | 0.20 | 0.01 | 0.00 | 0.00 | 1.00 | 0.18 |
| Gemini-2.5 Pro | 1 | 0.99 | **1.00** | 0.99 | 0.95 | 0.81 | 0.19 | **1.00** | **0.72** | 0.93 | **0.66** | **1.00** | **0.81** |
| | 2 | **1.00** | **1.00** | **0.99** | 0.18 | **0.66** | 0.13 | **1.00** | **0.36** | 0.24 | **0.49** | **1.00** | **0.66** |
| | 3 | **1.00** | **1.00** | 0.93 | 0.03 | 0.36 | **0.07** | 0.74 | 0.19 | 0.16 | 0.00 | **1.00** | 0.36 |
| Gemini-2.5 Flash | 1 | 0.83 | **1.00** | 0.69 | 0.86 | 0.64 | 0.19 | **1.00** | 0.30 | 0.81 | 0.19 | **1.00** | 0.75 |
| | 2 | 0.71 | 0.79 | 0.38 | 0.15 | 0.24 | 0.05 | **1.00** | 0.10 | 0.13 | 0.00 | **1.00** | 0.15 |
| | 3 | 0.50 | 0.53 | 0.21 | 0.01 | 0.22 | 0.03 | 0.58 | 0.06 | 0.05 | 0.01 | **1.00** | 0.11 |
| Claude-3.7 Sonnet | 1 | 0.98 | 0.97 | 0.85 | 0.64 | 0.57 | 0.22 | 0.97 | 0.54 | 0.60 | 0.00 | 0.98 | 0.43 |
| | 2 | 0.92 | 0.68 | 0.71 | 0.15 | 0.32 | **0.14** | 0.63 | 0.33 | 0.04 | 0.00 | 0.98 | 0.23 |
| | 3 | 0.81 | 0.51 | 0.37 | 0.05 | 0.18 | 0.01 | 0.44 | **0.27** | 0.01 | 0.00 | 0.99 | 0.10 |
| *Open-Source Models (Large)* | | | | | | | | | | | | | |
| QwenVL-2.5 (72B) | 1 | 0.48 | 0.98 | 0.68 | 0.42 | 0.41 | 0.29 | 0.62 | 0.00 | 0.65 | 0.12 | 0.96 | 0.38 |
| | 2 | 0.29 | 0.84 | 0.28 | 0.18 | 0.24 | 0.08 | 0.27 | 0.00 | 0.17 | 0.00 | 0.93 | 0.18 |
| | 3 | 0.01 | 0.65 | 0.09 | 0.03 | 0.09 | 0.04 | 0.15 | 0.00 | 0.00 | 0.00 | 0.92 | 0.08 |
| InternVL-3 (78B) | 1 | 0.43 | 0.99 | 0.59 | 0.47 | 0.48 | 0.26 | 0.63 | 0.01 | 0.48 | 0.00 | 0.91 | 0.41 |
| | 2 | 0.20 | 0.48 | 0.29 | 0.03 | 0.22 | 0.10 | 0.15 | 0.01 | 0.11 | 0.00 | 0.83 | 0.13 |
| | 3 | 0.05 | 0.17 | 0.09 | 0.01 | 0.08 | 0.06 | 0.09 | 0.02 | 0.00 | 0.00 | 0.75 | 0.05 |
| *Open-Source Models (Middle)* | | | | | | | | | | | | | |
| QwenVL-2.5 (32B) | 1 | 0.64 | **1.00** | 0.62 | 0.91 | 0.60 | 0.29 | 0.67 | 0.00 | 0.68 | 0.15 | 0.94 | 0.61 |
| | 2 | 0.35 | 0.81 | 0.44 | 0.15 | 0.30 | 0.04 | 0.33 | 0.02 | 0.09 | 0.00 | 0.92 | 0.19 |
| | 3 | 0.05 | 0.52 | 0.11 | 0.03 | 0.09 | 0.06 | 0.20 | 0.00 | 0.00 | 0.00 | 0.94 | 0.08 |
| InternVL-3 (38B) | 1 | 0.45 | 0.88 | 0.46 | 0.85 | 0.58 | 0.28 | 0.43 | 0.00 | 0.38 | 0.01 | 0.85 | 0.57 |
| | 2 | 0.22 | 0.40 | 0.29 | 0.35 | 0.27 | 0.09 | 0.25 | 0.00 | 0.12 | 0.00 | 0.76 | 0.32 |
| | 3 | 0.24 | 0.21 | 0.17 | 0.03 | 0.12 | 0.03 | 0.18 | 0.03 | 0.02 | 0.00 | 0.65 | 0.08 |
| *Open-Source Models (Small)* | | | | | | | | | | | | | |
| QwenVL-2.5 (7B) | 1 | 0.23 | 0.79 | 0.42 | 0.00 | 0.34 | 0.24 | 0.21 | 0.00 | 0.24 | 0.00 | 0.25 | 0.31 |
| | 2 | 0.07 | 0.31 | 0.20 | 0.00 | 0.16 | 0.07 | 0.10 | 0.00 | 0.04 | 0.00 | 0.20 | 0.11 |
| | 3 | 0.01 | 0.16 | 0.07 | 0.00 | 0.07 | 0.05 | 0.11 | 0.00 | 0.01 | 0.00 | 0.18 | 0.05 |
| InternVL-3 (8B) | 1 | 0.23 | 0.41 | 0.51 | 0.06 | 0.19 | 0.18 | 0.33 | 0.00 | 0.30 | 0.01 | 0.39 | 0.31 |
| | 2 | 0.05 | 0.11 | 0.26 | 0.02 | 0.13 | 0.09 | 0.19 | 0.01 | 0.02 | 0.00 | 0.33 | 0.14 |
| | 3 | 0.02 | 0.03 | 0.06 | 0.00 | 0.06 | 0.04 | 0.09 | 0.04 | 0.00 | 0.00 | 0.25 | 0.03 |
| DeepSeekVL-2 | 1 | 0.18 | 0.47 | 0.51 | 0.34 | 0.33 | 0.24 | 0.25 | 0.12 | 0.26 | 0.03 | 0.34 | 0.27 |
| | 2 | 0.06 | 0.09 | 0.14 | 0.01 | 0.12 | 0.09 | 0.12 | 0.04 | 0.03 | 0.07 | 0.25 | 0.10 |
| | 3 | 0.01 | 0.03 | 0.04 | 0.00 | 0.04 | 0.04 | 0.12 | 0.06 | 0.01 | **0.08** | 0.17 | 0.04 |
| *Human Baseline* | | | | | | | | | | | | | |
| Human | 1 | 0.98 | 1.00 | 1.00 | 1.00 | 1.00 | 1.00 | 0.97 | 1.00 | 1.00 | 1.00 | 1.00 | 1.00 |
| | 2 | 0.95 | 1.00 | 1.00 | 0.98 | 1.00 | 1.00 | 0.95 | 1.00 | 0.97 | 1.00 | 0.95 | 1.00 |
| | 3 | 0.93 | 1.00 | 1.00 | 0.97 | 1.00 | 1.00 | 0.93 | 1.00 | 0.95 | 1.00 | 0.95 | 1.00 |
| *Random Choose* | | | | | | | | | | | | | |
| Random | 1 | 0.24 | 0.25 | 0.25 | 0.38 | 0.25 | 0.25 | 0.05 | 0.25 | 0.50 | 0.25 | 0.25 | 0.15 |
| | 2 | 0.04 | 0.07 | 0.17 | 0.16 | 0.08 | 0.08 | 0.00 | 0.17 | 0.08 | 0.08 | 0.13 | 0.05 |
| | 3 | 0.02 | 0.02 | 0.13 | 0.12 | 0.04 | 0.04 | 0.00 | 0.13 | 0.04 | 0.04 | 0.13 | 0.03 |

## 6.4 CAPABILITY RADAR MAP

To provide deeper insights into the capabilities of MLLMs, as discussed in Section 3, we calculated capability scores across 5 dimensions and visualized them through radar map for each MLLM (Figure 3). The specific calculation methodology and formulas are detailed in Appendix D.

In **KIDGYM**, execution is a shared prerequisite: every task ultimately requires the model to translate its inferred goal into concrete, correct actions. To keep the other capability scores interpretable, we therefore measure execution explicitly only with the Classification (CL) task. In the remaining tasks, execution is still unavoidable, but the scoring is designed to emphasize the task's target capability rather than re-estimating execution in a more confounded setting. In practice, the CL score provides the clearest reference for execution readiness: models with weaker CL performance are more likely to accumulate action-level errors that depress performance and reduce the reliability of their scores on other capabilities. Notably, results suggest that closed-source models generally exhibit strong execution, whereas open-source models tend to lag, which can in turn degrade both their performance and the trustworthiness of their evaluations on other tasks.

Overall, closed-source models perform relatively well in learning and memory capabilities, yet a substantial gap compared with human performance remains. For open-source MLLMs, the performance of models in the same category generally improves as the number of parameters increases. However, there remains a significant gap between open-source and closed-source MLLMs, with o3, GPT-5 and Gemini-2.5-Pro standing out as particularly remarkable, dominating across all measured dimensions. As shown in the capability radar map, all evaluated MLLMs generally score lower in perception reasoning and planning capabilities. While these models have progressed beyond basic recognition tasks, they still struggle with more complex forms of visual cognition, particularly abstract and non-semantic ones. Similarly, the planning dimension requires further development to enable models to systematically organize tasks, predict the consequences of actions, and implement multi-step strategies for solving complex and composite problems.

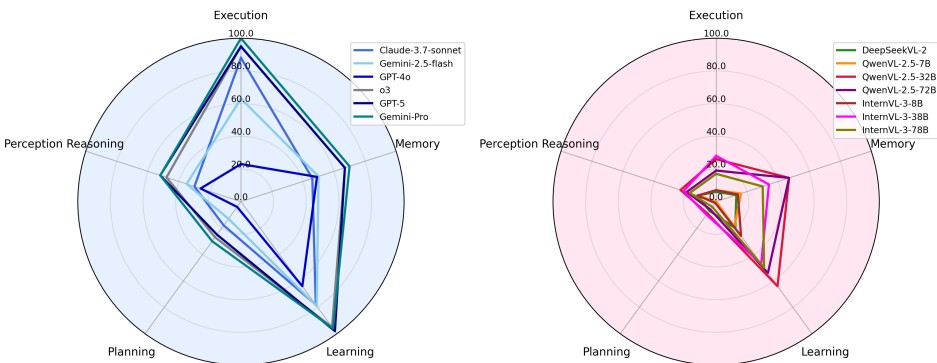

Figure 3: Five-dimensional capability radar chart. The chart on the left shows the capability scores of the closed-source models, while the chart on the right shows those of the open-source models.

## 7 CONCLUSION

In this work, we propose an assessment framework for MLLMs, incorporating five core capabilities based on the Wechsler Intelligence Scale. And we introduce KidGym, a comprehensive 2D grid-based benchmark for evaluating these capabilities of MLLMs. Experiments indicate that although some closed-source MLLMs can achieve a relatively high success rate on some simple tasks, they still exhibit obvious deficiencies in those compound tasks that require multiple capabilities. Specifically, more effort needs to be devoted to further enhancing models' abilities to handle non-semantic and quantitative visual information. Although the current number of tasks is limited, we believe that our open-source and scalable framework, KidGym, offers vast potential for the MLLM community and will drive further advancements in the field of AGI.

# 8 ACKNOWLEDGEMENT

This work is supported by Shanghai Sailing Program (23YF1427600).

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
