APPENDIX

# A  BENCHMARK DESIGN OVERVIEW

## A.1  WECHSLER INTELLIGENCE SCALE AND EXECUTIVE FUNCTION

Our work is chiefly informed by the Wechsler Intelligence Scales, specifically the fourth edition of the Wechsler Preschool and Primary Scale of Intelligence (WPPSI–IV) (Willis et al., 2015). Administered to children aged 2 years 5 months to 6 years 11 months, the WPPSI–IV is the most widely used assessment of cognitive ability worldwide. The Wechsler scales are periodically revised to maintain validity and practical relevance, with the WPPSI–IV representing the latest update. Wechsler conceptualized intelligence as a composite of multiple, quantitatively varying abilities. Accordingly, it reports scores across 5 core domains: Verbal Comprehension (**VCI**), Visual–Spatial (**VSI**), Fluid Reasoning (**FRI**), Working Memory (**WMI**), and Processing Speed (**PSI**).

Another important concept used in this study is Executive Function Diamond (2013) (**EF**). It refers to the general control mechanism by which a child conducts cognitive coordination when completing complex tasks to ensure that the cognitive system achieves specific goals in a flexible and optimized way. EF begins to develop in individuals at the age of 3 to 4 and gradually improves after adolescence. It has an important influence on a child's studies, life and sense of happiness.

## A.2  DESIGN PRINCIPLE FOR CAPACITY MAPPING

A fundamental issue in cognitive measurement known as the **"task impurity problem"**: no single task can purely measure one cognitive capacity in isolation. Any real-world task inevitably engages multiple cognitive processes simultaneously, and this is true whether we are evaluating human children or artificial agents (Miyake et al., 2000). Given this context, **KIDGYM** follows standard practice in psychometrics by designing tasks in which target abilities are dominant by design.

**Single-Capacity** tasks are designed such that:

- Performance is primarily driven by variation along one target capacity dimension.

- Non-target capabilities are explicitly minimized through mechanisms (e.g., keeping information visible to reduce memory load, or providing direct action options to reduce planning depth).

**Composite-Capacity** tasks are designed such that:

- Success requires coordination of multiple capacities that cannot be suppressed through mechanisms (e.g., remembering hidden information while performing sequential actions).

In addition, while Wechsler framed human intelligence as a constellation of interrelated abilities that vary in degree, the cognitive profile of MLLMs cannot be mapped one-to-one onto these human constructs. Consequently, each MLLM capability requires a definition that respects the model's architectural and functional particularities. Here's an explanation of how each MLLM capability discussed in the Section 3 connects to the specific cognitive domains measured in Appendix A.1.

**Execution** in MLLMs parallels the Processing Speed (PSI) of the WPPSI, which gauges how quickly and accurately children complete simple tasks, and it also maps onto key elements of Executive Function (EF). In MLLMs, however, solution time is heavily influenced by hardware and parameter size, making it a weak proxy for the model's reasoning capacity. Task-completion accuracy, by contrast, provides a far more informative measure. Accordingly, we define Execution as the model's capacity to carry out simple instructions in line with specific goals and constraints, producing outcomes that are both correct and precise.

**Perception Reasoning** in MLLMs corresponds to the WPPSI's Visual-Spatial (VSI), which measures a child's ability to interpret and organize visual information - and FRI, which assesses problem-solving and abstract thinking. We merge these two dimensions into a single construct, Perception Reasoning, defined as the model's capacity to interpret input data and draw logical inferences, thereby enabling it to grasp the implicit logical relationships across multimodal sources.

**Memory** in MLLMs parallels the WPPSI's Working Memory (WMI), which assesses a child's ability to hold and manipulate information over short intervals. For MLLMs, this dimension encompasses retaining contextual cues across interactions and retrieving relevant information when needed. Accordingly, we define Memory as the model's capacity to grasp context and latent relationships across multimodal inputs, ensuring responses that remain coherent and contextually appropriate.

**Learning** in MLLMs is grounded in the WPPSI's Verbal Comprehension Index (VCI), which gauges a child's ability to understand and apply new knowledge. Considering that any training corpus is inherently finite, progress toward AGI requires a model to keep acquiring knowledge beyond its initial dataset. Accordingly, we define Learning as MLLMs' capacity — anchored in deep linguistic and graphic understanding — to continually absorb, internalise, and adapt to new information.

**Planning** in MLLMs parallels Executive Function (EF) as well as the WPPSI's Visual Spatial Index (VSI) and Processing Speed Index (PSI), all of which support performance on complex tasks. We define Planning as the model's ability to strategize and sequence actions to achieve specific objectives while evaluating potential outcomes. This capability enables MLLMs to generate coherent, goal-oriented responses or actions, exhibiting higher-order thinking skills comparable to those that enhance children's performance across diverse cognitive tests.

### A.3 KIDGYM'S CONNECTIONS WITH WECHSLER

**Why Wechsler cannot be copied directly at the ability and experimental levels?**

Humans and MLLMs differ fundamentally in their embodiment and interaction modalities, so a literal transfer of abilities and subtests is inappropriate.

At the "ability" level, for example, the Processing Speed Index (PSI) in Wechsler is intended to capture individual differences in cognitive processing speed and sustained attention under time pressure. In the MLLM setting, however, inference latency is dominated by implementation factors (model size, quantization, batching, hardware accelerators, etc.). Treating wall-clock speed as an analog of PSI would therefore conflate engineering with cognitive ability and would not be scientifically meaningful. Similarly, the Working Memory Index (WMI) in Wechsler is designed to measure the ability to temporarily retain a limited amount of information —— content that may be difficult for humans to keep in mind, but relatively easy to operate on once maintained. In contrast, for MLLMs, previously seen images or tokens are typically stored in context buffers, so retaining the information itself is not the main challenge; instead, the difficulty lies in appropriately integrating and using that information within the current context.

At the "experimental" level, many Wechsler subtests rely on embodied, sensorimotor interactions that cannot be reproduced for MLLMs. For instance, certain tasks for young children require pointing to or touching parts of their own body, manipulating physical blocks, or drawing symbols by hand. These modalities (proprioception, handwriting) simply do not suit current MLLMs.

**How do we actually use the Wechsler framework?**

Given these limitations, we do not simply copy or fully implement the Wechsler test. Instead, we take the Wechsler framework as a design premise, since the Wechsler reflects nearly a century of knowledge about how to operationalize human cognitive abilities into measurable constructs. Although MLLMs are clearly not human, they are increasingly deployed in tasks that require heterogeneous cognitive abilities. By borrowing this conceptual structure, KIDGYM organizes the capabilities that MLLMs need to complete real-world tasks into interpretable dimensions.

Across the entire benchmark development process, we collaborated with our co-authors, who are experts in child brain science, to formulate a rigorous methodology that informed the design of the tasks. We first distilled each Wechsler index score into an underlying cognitive ability that is meaningful for MLLMs, and then constructed task families that systematically operationalize these abilities within a unified environment. When a Wechsler subtest could be transferred to an agent setting in an appropriate way, we applied only minimal modifications (e.g., replacing paper-and-pencil responses with discrete actions) while preserving the core cognitive demands. For subtests whose original format is not suitable for MLLMs, we co-designed new tasks with the expert, retaining the intended cognitive requirements while recasting the format to be maximally appropriate

for MLLMs. This collaborative and methodologically grounded procedure ensures that each ability dimension is supported by a coherent and well-specified set of cognitive requirements.

**Example: "Execution" and its relation to PSI**

In Wechsler, PSI tasks require children to perform simple, rule-based operations quickly and accurately. The core demand is to follow explicit instructions reliably under time constraints.

In **KIDGYM**, Execution tasks require MLLMs to accurately follow simple, explicitly specified rules in a structured environment (e.g., correctly categorizing objects). The core demand is instruction following and accurate action selection.

We operationalize this by evaluating success rate within a fixed step budget, which captures whether the model can efficiently complete tasks without unnecessary errors or redundant actions. This preserves the conceptual essence of PSI — "fast and accurate performance on relatively simple, well-defined tasks" — while adapting the measurement approach to the MLLM context.

# B  EXPERIMENT DETAILS

## B.1  RESOLUTION

The environment is a $9 \times 9$ grid with $64 \times 64$ pixel cells (total $576 \times 576$ pixels); gameplay occurs in a centered $5 \times 5$ region within an $8 \times 7$ background, and the remaining cells are decorative for semantic context. Interface elements are fixed: the bottom $8 \times 1$ strip is the backpack, and the left $9 \times 2$ block shows task hints. We adopt 64 pixels per grid — consistent with prior game-like evaluations (Hafner, 2021; Guss et al., 2019) — to balance clarity and compute at our scale (10800 questions per model across zero-shot, CoT, and ICL).

To assess resolution sensitivity, we additionally evaluated inputs at 32 and 96 pixels per grid:

Table 3: Resolution Experiment on Counting (CO) Task

|  | $32 \times 32$ | $96 \times 96$ | $96 \times 96$ |
|---|---|---|---|
| **o3** | 0.27 | 0.30 | 0.59 |
| **GPT-4o** | 0.00 | 0.00 | 0.00 |
| **Claude-3.7-Sonnet** | 0.14 | 0.54 | 0.79 |
| **DeepSeekVL-2** | 0.12 | 0.12 | 0.17 |
| **QwenVL-2.5(7B)** | 0.00 | 0.00 | 0.00 |
| **QwenVL-2.5(72B)** | 0.00 | 0.00 | 0.00 |

The results indicate that increasing the input resolution improves performance for some models on CO task, possibly because the higher resolution enlarges gaps between compact objects, enabling clearer separation and more effective divide-and-conquer counting. By contrast, humans maintain strong performance at 64 pixels, suggesting that the images already preserve sufficient detail to solve the task. This points to a remaining gap in visual processing efficiency for MLLMs relative to humans. When the resolution was reduced to 32 pixels, the loss of certain details led to a significant drop in the accuracy of most MLLMs.

## B.2  QUESTION NUMBER

**KIDGYM** comprises 12 distinct tasks, each implemented at three difficulty levels (Level 1–Level 3). For every ⟨task, level⟩ pair, we ran 100 randomly generated episodes. Thus, a model is evaluated on $12 \times 3 \times 100 = 3600$ questions across all tasks and difficulty levels. In addition to the zero-shot setting, we also evaluate CoT and ICL, yielding $3 \times 3{,}600 = 10800$ total evaluations.

More precisely, **KIDGYM** employs procedural generation: episodes are generated dynamically at runtime by sampling from a catalog space, so the questions are theoretically "**inexhaustible**".

To make this concrete, consider the Level-1 instance of the Classification (CL) task:

- On a 5×5 map (25 squares), place 1 agent, 2 items and 2 baskets.
- Items are selected from 4 categories (animals/fruits/food/toys), and the baskets come in 4 colors.

The number of distinct states for this single level is more than $10^{14}$ different states.

During generation, **KIDGYM** performs reachability checks to ensure every episode is solvable. Even with this constraint, each task still contains far more instances than the 100 we sample for evaluation. This breadth ensures that repeated evaluations of the same task present different questions (distinct scenario/location/item combinations) — thereby mitigating data contamination and enabling a more effective and accurate assessment of model capabilities.

### B.3 PARAMETERS

In this study, we set the temperature parameter of all language models to 0 for all experimental tasks. By doing so, we enforced deterministic behavior as possible, ensuring that the models' outputs were exclusively determined by their learned probability distributions. This configuration minimizes the influence of stochasticity and provides a controlled environment for evaluating model performance.

### B.4 ACTION LIST

In **KIDGYM**, each task has a function called "generate_actions()", before the agent selects its next move, the environment invokes this function to enumerate all currently executable actions. The resulting action list is determined by (i) the types of items present, (ii) their current states (e.g., whether an item is in the scene or in the backpack), and (iii) the number of items in the environment.

For example, in the Classification task we use the following rules:

**Type**: Collectible items (e.g., apple/egg/ball) are declared pickable and baskets are unpickable by default; therefore the system emits 'PickUp(item)' only for pickable objects.

**State**: Based on a pickable item's state, the system includes PickUp(item) when the item is in the scene and PutDown(item) once it has been collected into the backpack.

**Quantity**: At each step, the system instantiates all applicable actions for every item present — no matter how many — producing an exhaustive action list that is then randomly permuted.

This design enables MLLMs to flexibly complete tasks via multiple valid action sequences. For example, one can either pick up all items first and then place them into the basket one by one, or pick up and place items alternately.

### B.5 RANDOM BASELINE

To provide a benchmark for comparison, we established a random baseline in which actions were selected entirely at random. The results of the random baseline were obtained either through analytical computation or experimental estimation, depending on the complexity of the task. For simpler tasks, such as SE, FI, PU, PL, SO, MDE and MFI, probabilistic methods were used to compute the expected performance of random actions.

For more complex tasks, such as CL, MA, CO, DMA and MMA, where analytical solutions are impractical, the random baseline was approximated by running 500 iterations of random experiments. This methodology allows the random baseline to serve as a meaningful point of reference across a diverse range of tasks.

### B.6 EVALUATION PROCEDURE

Our evaluation process is illustrated in Algorithm 1.

---

**Algorithm 1** Model Evaluation

---

 1: **Input:** $env$: task environment, $model$: MLLM model, $history$: conversation history
 2: **while** True **do**
 3:     $prompt \leftarrow$ **GeneratePrompt**$(env)$
 4:     $response, history \leftarrow$ **Chat**$(model, prompt, history)$
 5:     $action \leftarrow$ **ProcessAnswer**$(response)$
 6:     **Step**$(env, action)$
 7:     **if** (Task Over) **or** (Reach Max Steps) **then**
 8:         **exit the while loop**
 9:     **end if**
10: **end while**

---

For each task, we first generate prompts containing the basic rules and task descriptions. Next, the MLLMs receive prompts and conversation history to generate output. Finally, we process and analyze the various responses generated. The specific algorithm for answer analysis can be referred to in Appendix B.8.

### B.7 PROMPT DESIGN

To ensure fairness across all MLLMs, we conducted experiments using exactly the same prompts that included all instruction rules. The sole variation lay in the input context: for tasks that did not test memory, the model received only an image representing the current environmental state, whereas for memory-dependent tasks it was given the entire sequence of historical states.

For evaluations that did not require chain-of-thought (CoT) reasoning, the model was instructed to output only the chosen answer option, expressed as a capital letter. When CoT was required, we asked the model to include an explicit, step-by-step rationale alongside its answer.

For ICL evaluation, we furnish the MLLM with a fully worked task example that includes images of every intermediate state, the correct action at each step, and the corresponding reason. Detailed examples can be found in the Appendix G.

Prompts detailing each task's <GOAL> and <ACTIONS> are provided in the Appendix F. To avoid the impact caused by fixed options, we map the original high-dimensional action space to a set of randomized answer options, from which the MLLMs must select.

**Example**

The action list is: [*'pick up apple', 'pick up banana', 'pick up orange'*].

The action choice in prompt will be: *A) 'pick up orange', B) 'pick up apple', C) 'pick up banana'*.

### B.8 ANSWER DECODE

Even though we emphasized in the prompts that the MLLM should respond in the specified format of the options, we were still unable to strictly standardize their response formats for all MLLMs. Therefore, we collected a large number of responses to analyze and summarize the characteristics of MLLMs when answering our questions. The final decoding method was determined as follows.

**Explanation**

Since chain-of-thought prompts and some multimodal large language models may wrap their responses in '<answer >...</answer>' tags, we begin by stripping these optional tags so that only the raw answer text is processed. Then, we iterate through the list of actions and check whether any action appears in the response generated by the MLLM. If a match is found, the index of the matching action is returned immediately. If no match is found, we then search for the first single uppercase letter, and check whether its corresponding index falls within the valid range of the action list. If it does, the corresponding index is returned; otherwise, the response is considered invalid.

**Example**

*response*: *<answer >A </answer >* $\rightarrow return\,A$

*response*: $A \rightarrow return\,A$

*response*: *I choose action letter B) 'pick up item with label 2'.* $\rightarrow return\,B$

*response*: *Based on all of the information, I choose action C.* $\rightarrow return\,C$

*response*: *I'm sorry, but I can't provide the correct answer as the image does not contain a dog. It appears to be a game with various animals, but none of them are dogs.* $\rightarrow return\,NONE$

*response*: *...?-=\== ..n\n The-1\n\n The-1* $\rightarrow return\,NONE$

---

**Algorithm 2** Process Answer

---

1: **Input:** $answer$: string, $actions$: list of strings
2: $m \leftarrow$ regexSearch(``<answer>(.*?)</answer>'', $answer$)
3: **if** $m \neq$ None **then**
4:     $answer \leftarrow m$.group(1)
5: **end if**
6: **for** each action in $actions$ **do**
7:     **if** action is in $answer$ **then**
8:       **return** index of action
9:     **end if**
10: **end for**
11: $match \leftarrow$ first uppercase letter found in $answer$
12: **if** $match$ is not None and the index of $match$ in alphabet is less than length of actions **then**
13:     $index \leftarrow$ index of $match$ in alphabet
14:     **return** $index$
15: **end if**
16: **return** None

---

---

**Prompts for KidGym's Tasks**

*You are playing as a character in a 2D game scene – the man dressed in a brown shirt and gray pants. During the game, you need to complete a task step by step by interacting with various items within the scene.*

*The following are the rules that you need to understand and follow:*
*- Items in the scene are identified by numerical labels, such as 0, 1, 2, 3, etc.*
*- At the bottom of the scene, there are four black squares representing your backpack slots, labeled A, B, C, and D.*
*- You can directly interact with items in the scene (e.g., picking up an apple) or use items from your backpack to interact with other items. (e.g., unlocking a door with a key).*

[If **ICL** is required]
*To help you better understand the rules, let's go through an example step by step.*
`<EXAMPLE>`

[If **Memory** capability is **NOT** required]
*Each step, I will provide you with a snapshot of the current state, along with a list of actionable options that you can choose to perform at this stage. Then you should analyze the given information and select the correct action to achieve the goal.*

[Else: **Memory** capability **IS** required]
*Each step, I will provide you with a snapshot of the current state as well as a collection of all the previous states of the scene, along with a list of actionable options that you can choose to perform at this stage. Then you should analyze the given information and select the correct action to achieve the goal.*

*In this task, your goal is: `<GOAL>`. Now the game starts!*
*What is the action you will choose? The actions you can choose from are: `<ACTIONS>`.*

[If **CoT** is **NOT** required]
*Please directly give your answer within `<ANSWER></ANSWER>` tags.*
*i.e., `<ANSWER>` answer here `</ANSWER>`*
*The answer should only contain the uppercase letter.*

[Else: **CoT IS** required]
*You should first think about the reasoning process in your mind and then provide the answer. The reasoning process and answer are enclosed within `<THINK></THINK>` and `<ANSWER></ANSWER>` tags, respectively.*
*i.e., `<THINK>` reasoning process here `</THINK>` `<ANSWER>` answer here `</ANSWER>`.*
*The answer should only contain the uppercase letter.*

## C  HUMAN BASELINE

We conducted human testing with participants aged 20–25, spanning university students from first-year undergraduates to first-year graduate students. For each task and difficulty level, at least 30 participants completed an online test identical in format to the MLLM evaluation. The results are reported in Table 4 and Table 5.

Table 4: Human performance results on tasks CL, SE, SO, MA, FI, PU.

|     | CL | SE | SO | MA | FI | PU |
|-----|-----|-----|-----|-----|-----|-----|
| L1 | 0.98 (39/40) | 1.00 (38/38) | 1.00 (45/45) | 1.00 (44/44) | 1.00 (41/41) | 1.00 (39/39) |
| L2 | 0.95 (37/39) | 1.00 (41/41) | 0.97 (36/37) | 0.98 (39/40) | 1.00 (38/38) | 1.00 (43/43) |
| L3 | 0.93 (40/43) | 1.00 (43/43) | 0.95 (38/40) | 0.97 (37/38) | 1.00 (43/43) | 1.00 (40/40) |

Table 5: Human performance results on tasks PL, CO, DMA, MMA, MDE, MFI.

|     | PL | CO | DMA | MMA | MDE | MFI |
|-----|-----|-----|-----|-----|-----|-----|
| L1 | 1.00 (43/43) | 1.00 (37/37) | 1.00(41/41) | 0.97 (37/38) | 1.00 (39/39) | 1.00 (41/41) |
| L2 | 1.00 (38/38) | 1.00 (44/44) | 1.00(40/40) | 0.95 (37/39) | 1.00 (41/41) | 1.00 (43/43) |
| L3 | 0.95 (39/41) | 1.00 (41/41) | 1.00(41/41) | 0.92 (42/45) | 1.00 (42/42) | 1.00 (38/38) |

As shown in the table, almost all tasks achieved near-perfect accuracy, with the lowest score being 0.93. We conducted interviews with the participants, and they reported that some errors were due to accidental misclicks during the online test and all tasks were relatively easy for them to complete. Therefore, using accuracy (with a maximum of 1.00) as a measure of the model's performance is reasonable, as it clearly reflects the gap between the model and human performance.

## D  CAPABILITY SCORE

To provide further understanding of the individual agent capabilities of MLLM, as discussed in Section 3, we calculated the capability scores and generate a five-dimensional radar chart for each MLLM (see in Figure 3), the experiment result can be found in Table 6.

For each MLLM, we first compute a task-level weighted success rate:

$$w_t \; = \; 0.2\, p_{t,1} \; + \; 0.3\, p_{t,2} \; + \; 0.5\, p_{t,3},$$

where $p_{t,d}$ is the model's success rate on task $t$ at difficulty level $d \in \{1, 2, 3\}$. Next, the score for a capability $c$ is obtained by aggregating these weighted success rates over all tasks $t_i$ associated with that capability, with $T_c$ denoting the set of tasks that measure capability $c$.

$$\text{Score}(c) \; = \; 100 \times \frac{1}{|T_c|} \sum_{t_i \in T_c} w_{t_i},$$

## E  CoT AND ICL RESULTS

To systematically compare the zero-shot, CoT, and ICL reasoning paradigms, we selected four representative tasks and summarized their strengths and limitations:

**Chain-of-Thought (CoT)**: By comparing the performance of the three tasks under zero-shot and CoT (CL/PL/SO), we find that CoT significantly boosts performance on tasks that emphasize action-level operations, whose core requirements are preciese instruction following and accurate action selection. In such settings, step-by-step reasoning encourages the model to explicitly consider the rationale behind each intermediate step, thereby directly reducing execution errors.

Table 6: Comparison of MLLM's capability scores.

| | Execution | Memory | Learning | Planning | Perception Reasoning |
|---|---|---|---|---|---|
| *Closed-Source Models (API)* | | | | | |
| o3 | 95 | 67 | 80 | 30 | 43 |
| GPT-5 | 95 | 67 | **98** | 30 | 46 |
| GPT-4o | 23 | 49 | 43 | 7 | 21 |
| Gemini-2.5-pro | **100** | **70** | 79 | **31** | **48** |
| Gemini-2.5-flash | 63 | 50 | 59 | 15 | 31 |
| Claude-3.7-sonnet | 88 | 46 | 57 | 17 | 31 |
| *Open-Source Models (Large)* | | | | | |
| QwenVL-2.5(72B) | 19 | 47 | 41 | 9 | 15 |
| InternVL-3(78B) | 17 | 34 | 34 | 5 | 14 |
| *Open-Source Models (Middle)* | | | | | |
| QwenVL-2.5(32B) | 26 | 47 | 44 | 11 | 18 |
| InternVL-3(38B) | 28 | 34 | 34 | 11 | 17 |
| *Open-Source Models (Small)* | | | | | |
| QwenVL-2.5(7B) | 7 | 16 | 14 | 2 | 10 |
| InternVL-3(8B) | 7 | 14 | 19 | 3 | 10 |
| DeepSeekVL-2 | 6 | 13 | 15 | 7 | 11 |
| *Average Score(Models)* | | | | | |
| Average | 38 | 40 | 57 | 10 | 20 |
| *Human Score* | | | | | |
| Human | 96 | 99 | 99 | 97 | 100 |

**In-context Learning (ICL)**: We find that ICL is not uniformly advantageous and can even underperform zero-shot on certain capability types. In particular, for tasks that emphasize memory (e.g., SE) and learning/adaptation (e.g., SO), models sometimes overfit to the provided examples and underweight the actual attributes of the current scene. In these cases, the model appears to "replicate" patterns from the demonstrations rather than dynamically integrating new environmental cues, which can harm generalization. This suggests that when the task type remains the same but the specific scenes change, ICL may be less effective than a simple zero-shot strategy.

Table 7: Comparison of MLLM performances across 12 **KIDGYM** tasks in CoT.

| Methods | L | CL | SE | SO | MA | FI | PU | PL | CO | DMA | MMA | MDE | MFI |
|---------|---|------|------|------|------|------|------|------|------|------|------|------|------|
| *Closed-Source Models (API)* | | | | | | | | | | | | | |
| o3 | 1 | 1.00 | 1.00 | 0.97 | 0.95 | 0.81 | 0.24 | 1.00 | 0.25 | 0.92 | 0.47 | 0.99 | 0.81 |
|  | 2 | 0.98 | 1.00 | 0.97 | 0.70 | 0.63 | 0.12 | 0.98 | 0.25 | 0.51 | 0.50 | 1.00 | 0.52 |
|  | 3 | 0.92 | 1.00 | 0.89 | 0.73 | 0.31 | 0.03 | 0.81 | 0.12 | 0.20 | 0.71 | 1.00 | 0.35 |
| GPT-4o | 1 | 0.95 | 1.00 | 0.83 | 0.80 | 0.58 | 0.25 | 0.95 | 0.05 | 0.68 | 0.16 | 0.97 | 0.64 |
|  | 2 | 0.96 | 0.92 | 0.59 | 0.31 | 0.37 | 0.02 | 0.38 | 0.03 | 0.10 | 0.01 | 1.00 | 0.27 |
|  | 3 | 0.24 | 0.82 | 0.26 | 0.23 | 0.12 | 0.03 | 0.24 | 0.04 | 0.02 | 0.34 | 0.99 | 0.15 |
| Gemini-2.5 Flush | 1 | 0.86 | 1.00 | 0.84 | 0.83 | 0.70 | 0.24 | 1.00 | 0.37 | 0.61 | 0.28 | 1.00 | 0.83 |
|  | 2 | 0.62 | 0.84 | 0.43 | 0.15 | 0.32 | 0.05 | 0.98 | 0.14 | 0.26 | 0.03 | 1.00 | 0.18 |
|  | 3 | 0.40 | 0.53 | 0.27 | 0.04 | 0.17 | 0.06 | 0.67 | 0.10 | 0.09 | 0.01 | 1.00 | 0.10 |
| Claude-3.7 Sonnet | 1 | 0.98 | 0.97 | 0.98 | 0.56 | 0.61 | 0.31 | 0.97 | 0.59 | 0.33 | 0.04 | 0.98 | 0.42 |
|  | 2 | 0.83 | 0.81 | 0.84 | 0.44 | 0.32 | 0.05 | 0.81 | 0.43 | 0.05 | 0.00 | 0.97 | 0.23 |
|  | 3 | 0.93 | 0.73 | 0.46 | 0.43 | 0.19 | 0.03 | 0.57 | 0.33 | 0.05 | 0.10 | 1.00 | 0.14 |
| *Open-Source Models (Large)* | | | | | | | | | | | | | |
| QwenVL-2.5 (72B) | 1 | 0.93 | 1.00 | 0.78 | 0.69 | 0.37 | 0.26 | 0.84 | 0.02 | 0.78 | 0.19 | 0.97 | 0.26 |
|  | 2 | 0.70 | 0.93 | 0.56 | 0.37 | 0.28 | 0.09 | 0.34 | 0.02 | 0.19 | 0.02 | 0.98 | 0.20 |
|  | 3 | 0.23 | 0.67 | 0.25 | 0.15 | 0.11 | 0.06 | 0.19 | 0.01 | 0.00 | 0.00 | 0.97 | 0.05 |
| InternVL-3 (78B) | 1 | 0.86 | 1.00 | 0.68 | 0.76 | 0.55 | 0.30 | 0.84 | 0.10 | 0.58 | 0.13 | 0.94 | 0.54 |
|  | 2 | 0.66 | 0.88 | 0.62 | 0.18 | 0.22 | 0.06 | 0.33 | 0.03 | 0.06 | 0.03 | 0.93 | 0.25 |
|  | 3 | 0.30 | 0.62 | 0.44 | 0.07 | 0.18 | 0.01 | 0.14 | 0.01 | 0.01 | 0.01 | 0.84 | 0.08 |
| *Open-Source Models (Small)* | | | | | | | | | | | | | |
| QwenVL-2.5 (7B) | 1 | 0.31 | 0.86 | 0.50 | 0.00 | 0.25 | 0.27 | 0.35 | 0.00 | 0.27 | 0.00 | 0.79 | 0.26 |
|  | 2 | 0.09 | 0.37 | 0.26 | 0.07 | 0.07 | 0.08 | 0.11 | 0.02 | 0.04 | 0.03 | 0.81 | 0.08 |
|  | 3 | 0.04 | 0.18 | 0.09 | 0.04 | 0.04 | 0.05 | 0.07 | 0.02 | 0.02 | 0.02 | 0.77 | 0.05 |
| InternVL-3 (8B) | 1 | 0.31 | 0.65 | 0.55 | 0.18 | 0.22 | 0.21 | 0.29 | 0.03 | 0.27 | 0.01 | 0.58 | 0.25 |
|  | 2 | 0.08 | 0.12 | 0.42 | 0.03 | 0.14 | 0.10 | 0.14 | 0.02 | 0.08 | 0.00 | 0.54 | 0.14 |
|  | 3 | 0.05 | 0.03 | 0.08 | 0.01 | 0.07 | 0.02 | 0.11 | 0.01 | 0.01 | 0.00 | 0.44 | 0.04 |
| DeepSeekVL-2 | 1 | 0.26 | 0.46 | 0.48 | 0.40 | 0.33 | 0.27 | 0.28 | 0.16 | 0.18 | 0.03 | 0.34 | 0.29 |
|  | 2 | 0.06 | 0.11 | 0.16 | 0.21 | 0.09 | 0.11 | 0.13 | 0.04 | 0.03 | 0.07 | 0.34 | 0.09 |
|  | 3 | 0.02 | 0.02 | 0.04 | 0.13 | 0.06 | 0.04 | 0.12 | 0.04 | 0.00 | 0.07 | 0.27 | 0.04 |
| *Random Choose* | | | | | | | | | | | | | |
| Random | 1 | 0.24 | 0.25 | 0.50 | 0.38 | 0.25 | 0.25 | 0.25 | 0.15 | 0.50 | 0.05 | 0.25 | 0.25 |
|  | 2 | 0.04 | 0.07 | 0.08 | 0.16 | 0.08 | 0.08 | 0.13 | 0.05 | 0.08 | 0.00 | 0.17 | 0.08 |
|  | 3 | 0.02 | 0.02 | 0.04 | 0.12 | 0.04 | 0.04 | 0.13 | 0.03 | 0.14 | 0.00 | 0.13 | 0.04 |

Table 8: Comparison of MLLM performances across 12 **KIDGYM** tasks in ICL.

| Methods | L | CL | SE | SO | MA | FI | PU | PL | CO | DMA | MMA | MDE | MFI |
|---|---|---|---|---|---|---|---|---|---|---|---|---|---|
| *Closed-Source Models (API)* | | | | | | | | | | | | | |
| o3 | 1 | 1.00 | 0.96 | 0.83 | 0.89 | 0.70 | 0.25 | 1.00 | 0.41 | 0.96 | 0.66 | 1.00 | 0.73 |
| | 2 | 0.98 | 1.00 | 0.88 | 0.77 | 0.55 | 0.06 | 0.99 | 0.16 | 0.59 | 0.63 | 1.00 | 0.56 |
| | 3 | 0.99 | 0.99 | 0.74 | 0.79 | 0.27 | 0.03 | 0.86 | 0.15 | 0.18 | 0.54 | 0.99 | 0.26 |
| GPT-4o | 1 | 0.32 | 0.80 | 0.46 | 0.38 | 0.32 | 0.21 | 0.62 | 0.00 | 0.30 | 0.08 | 0.54 | 0.36 |
| | 2 | 0.29 | 0.47 | 0.18 | 0.22 | 0.15 | 0.08 | 0.13 | 0.00 | 0.14 | 0.00 | 0.88 | 0.09 |
| | 3 | 0.09 | 0.24 | 0.05 | 0.10 | 0.08 | 0.05 | 0.20 | 0.00 | 0.01 | 0.06 | 0.91 | 0.07 |
| Gemini-2.5 Flush | 1 | 0.99 | 0.82 | 0.72 | 0.50 | 0.38 | 0.20 | 1.00 | 0.02 | 0.49 | 0.07 | 0.95 | 0.32 |
| | 2 | 0.73 | 0.83 | 0.44 | 0.09 | 0.30 | 0.07 | 0.99 | 0.01 | 0.03 | 0.01 | 0.90 | 0.25 |
| | 3 | 0.47 | 0.75 | 0.25 | 0.01 | 0.29 | 0.01 | 0.69 | 0.00 | 0.05 | 0.01 | 0.81 | 0.21 |
| Claude-3.7 Sonnet | 1 | 0.69 | 0.47 | 0.52 | 0.31 | 0.42 | 0.27 | 0.61 | 0.09 | 0.27 | 0.08 | 0.73 | 0.30 |
| | 2 | 0.60 | 0.40 | 0.51 | 0.32 | 0.24 | 0.09 | 0.40 | 0.03 | 0.07 | 0.00 | 0.88 | 0.13 |
| | 3 | 0.31 | 0.39 | 0.35 | 0.34 | 0.13 | 0.05 | 0.28 | 0.01 | 0.01 | 0.00 | 0.92 | 0.09 |
| *Open-Source Models (Large)* | | | | | | | | | | | | | |
| QwenVL-2.5 (72B) | 1 | 0.30 | 0.38 | 0.63 | 0.00 | 0.21 | 0.28 | 0.54 | 0.00 | 0.23 | 0.19 | 0.97 | 0.25 |
| | 2 | 0.01 | 0.23 | 0.18 | 0.00 | 0.06 | 0.09 | 0.25 | 0.00 | 0.03 | 0.02 | 0.98 | 0.06 |
| | 3 | 0.04 | 0.08 | 0.05 | 0.00 | 0.10 | 0.05 | 0.19 | 0.00 | 0.00 | 0.00 | 0.97 | 0.08 |
| InternVL-3 (78B) | 1 | 0.21 | 0.36 | 0.62 | 0.27 | 0.28 | 0.21 | 0.44 | 0.04 | 0.17 | 0.01 | 0.36 | 0.22 |
| | 2 | 0.08 | 0.29 | 0.34 | 0.03 | 0.10 | 0.06 | 0.14 | 0.01 | 0.05 | 0.00 | 0.34 | 0.11 |
| | 3 | 0.05 | 0.03 | 0.12 | 0.01 | 0.04 | 0.04 | 0.11 | 0.01 | 0.00 | 0.00 | 0.28 | 0.02 |
| *Open-Source Models (Small)* | | | | | | | | | | | | | |
| QwenVL-2.5 (7B) | 1 | 0.35 | 0.28 | 0.50 | 0.00 | 0.25 | 0.25 | 0.21 | 0.00 | 0.13 | 0.02 | 0.31 | 0.21 |
| | 2 | 0.03 | 0.12 | 0.18 | 0.04 | 0.08 | 0.07 | 0.08 | 0.01 | 0.02 | 0.02 | 0.18 | 0.07 |
| | 3 | 0.00 | 0.04 | 0.08 | 0.02 | 0.04 | 0.03 | 0.09 | 0.00 | 0.01 | 0.01 | 0.14 | 0.03 |
| InternVL-3 (8B) | 1 | 0.32 | 0.30 | 0.64 | 0.35 | 0.16 | 0.22 | 0.26 | 0.08 | 0.14 | 0.03 | 0.24 | 0.19 |
| | 2 | 0.09 | 0.05 | 0.27 | 0.03 | 0.05 | 0.07 | 0.14 | 0.05 | 0.04 | 0.00 | 0.21 | 0.12 |
| | 3 | 0.06 | 0.03 | 0.10 | 0.01 | 0.04 | 0.05 | 0.10 | 0.03 | 0.00 | 0.00 | 0.12 | 0.09 |
| DeepSeekVL-2 | 1 | 0.22 | 0.21 | 0.49 | 0.22 | 0.25 | 0.24 | 0.24 | 0.18 | 0.17 | 0.02 | 0.27 | 0.21 |
| | 2 | 0.06 | 0.07 | 0.15 | 0.16 | 0.09 | 0.09 | 0.12 | 0.06 | 0.02 | 0.03 | 0.15 | 0.08 |
| | 3 | 0.03 | 0.00 | 0.03 | 0.15 | 0.02 | 0.02 | 0.11 | 0.07 | 0.02 | 0.05 | 0.11 | 0.01 |
| *Random Choose* | | | | | | | | | | | | | |
| Random | 1 | 0.24 | 0.25 | 0.50 | 0.38 | 0.25 | 0.25 | 0.25 | 0.15 | 0.50 | 0.05 | 0.25 | 0.25 |
| | 2 | 0.04 | 0.07 | 0.08 | 0.16 | 0.08 | 0.08 | 0.13 | 0.05 | 0.08 | 0.00 | 0.17 | 0.08 |
| | 3 | 0.02 | 0.02 | 0.04 | 0.12 | 0.04 | 0.04 | 0.13 | 0.03 | 0.04 | 0.00 | 0.13 | 0.04 |

## F   TASK INFORMATION

### F.1   CLASSIFICATION (CL)

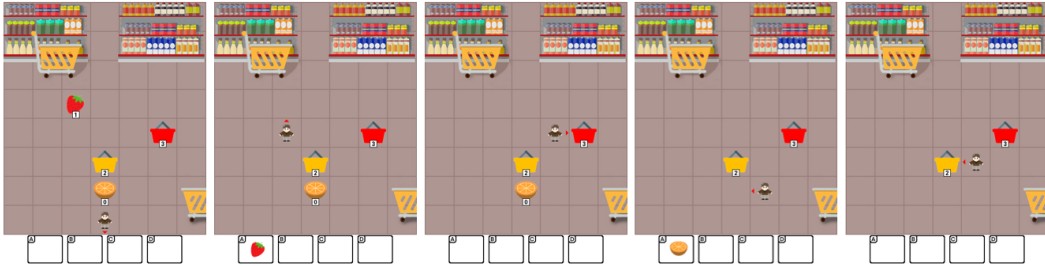

Figure 4: An example of task Classification (Level1). In this example, the goal is: "*Place strawberry in red basket and orange in yellow basket respectively.*".

**Introduction**

This task requires the agent to place two different items into containers with designated color according to given instructions.

**Goal**

Place $\langle ITEM_1 \rangle$ in $\langle CONT_1 \rangle$ and $\langle ITEM_2 \rangle$ in $\langle CONT_2 \rangle$ respectively.

**Actions**

pick up the item with label $\langle CONT.ID \rangle$
put the item from backpack $\langle BAG.ID \rangle$ into the basket with label $\langle CONT.ID \rangle$

**Difficulty Level**

Level1: There is **1** of each kind of item.
Level2: There are **2** of each kind of item.
Level3: There are **3** of each kind of item.

**Example** (see Figure 4)

- Step1
    - Action List
        * A) 'pick up item with label 1'
        * B) 'pick up item with label 0'
- Step2
    - Action List
        * A) 'put the item from backpack A into the basket with label 2'
        * B) 'put the item from backpack A into the basket with label 3'
        * C) 'pick up item with label 0'
- Step3
    - Action List
        * A) 'pick up item with label 0'
- Step4
    - Action List
        * A) 'put the item from backpack A into the basket with label 3'
        * B) 'put the item from backpack A into the basket with label 2'

## F.2 COUNTING (CO)

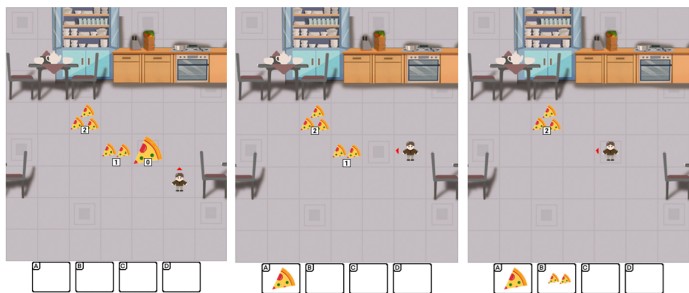

Figure 5: An example of task Counting (Level1). In this example, the goal is: "*Collect 3 pizzas.*".

**Introduction**

This task requires the agent to collect a certain number of items.

**Goal**

Collect $\langle NUM \rangle \langle ITEM \rangle$. Make sure you have gathered exactly this amount, no more and no less. You should be aware that there may be 1 to 3 items of different quantities in one grid. Once you have collected this number of $\langle ITEM \rangle$, select the action: "I have already collected $\langle NUM \rangle \langle ITEM \rangle$".

**Actions**

pick up $\langle ITEM \rangle$ with label $\langle ITEM.ID \rangle$
I have already collected $\langle NUM \rangle \langle ITEM \rangle$

**Difficulty Level**

Level1: The agent needs to collect **1-3** items.
Level2: The agent needs to collect **2-6** items.
Level3: The agent needs to collect **3-9** items.

**Example** (see Figure 5)

- Step1
  - Action List
    * A) 'pick up pizza with label 1'
    * B) 'pick up pizza with label 2'
    * C) 'pick up pizza with label 0'
    * D) 'I have already collected 3 pizzas'
- Step2
  - Action List
    * A) 'pick up pizza with label 2'
    * B) 'pick up pizza with label 1'
    * C) 'I have already collected 3 pizzas'
- Step3
  - Action List
    * A) 'I have already collected 3 pizzas'
    * B) 'pick up pizza with label 2'

## F.3 SELECTION (SE)

**Introduction**

This task requires the agent to first memorize the items and then collect all of them from the scene.

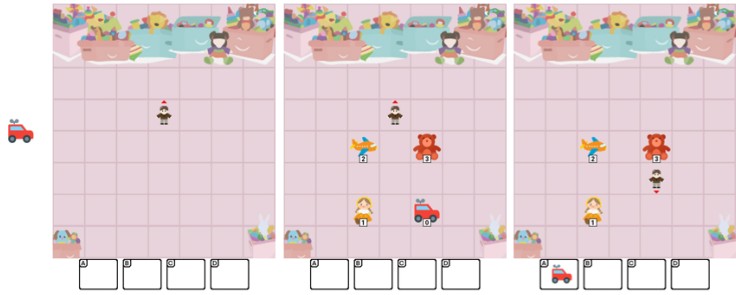

Figure 6: An example of task Selection (Level1). In this example, the goal is: "*Remember the toy car in the first image and then select it from the scene.*".

**Goal**

In the first image, an item will be shown on the left margin that you need to remember. In the following images, several random items will be generated in the scene, and you need to select the one you recall. If you understand the rules, select the 'continue' action to start the task.

**Actions**

choose $\langle ITEM \rangle$ with label $\langle ITEM.ID \rangle$

**Difficulty Level**

Level1: The agent needs to remember and select $\underline{\mathbf{1}}$ item.
Level2: The agent needs to remember and select $\underline{\mathbf{2}}$ items.
Level3: The agent needs to remember and select $\underline{\mathbf{3}}$ items.

**Example** (see Figure 6)

- Step1
  - Action List
    * A) 'conitnue'
- Step2
  - Action List
    * A) 'choose toy with label 3'
    * B) 'choose toy with label 1'
    * C) 'choose toy with label 0'
    * D) 'choose toy with label 2'

### F.4 MEMORY DECODE (MDE)

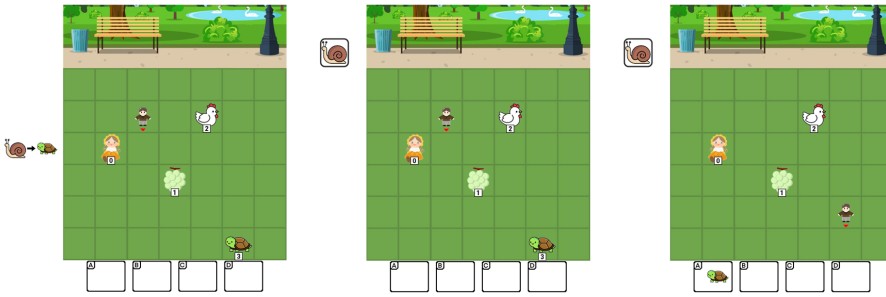

Figure 7: An example of task Memory Decode (Level1). In this example, the goal is: "*Remember and select the item corresponding to the snail in the scene based on the relationship.*".

**Introduction**

This task requires the agent not only to learn the association rules but also to remember them and choose the correct item.

**Goal**

In the first image, arrow-connected items with one-to-one correspondence(s) will be shown on the left margin that you need to remember. In the following images, the correspondence(s) will not be shown, and a target item will be generated in the black box in the upper left corner. You need to select the correct corresponding item for the target based on the pairing you remembered in the first image. If you understand the rules, choose 'continue' to begin the task.

**Actions**

choose $\langle ITEM \rangle$ with label $\langle ITEM.ID \rangle$

**Difficulty Level**

Level1: **1** set of correspondences is displayed in the first image.
Level2: **2** sets of correspondences are displayed in the first image.
Level3: **3** sets of correspondences are displayed in the first image.

**Example** (see Figure 7)

- Step1
  - Action List
    * A) 'continue'

- Step2
  - Action List
    * A) 'choose item with label 0'
    * B) 'choose item with label 2'
    * C) 'choose item with label 1'
    * D) 'choose item with label 3'

### F.5 PUZZLE (PU)

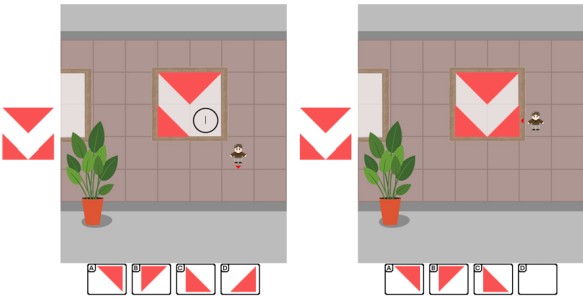

Figure 8: An example of task Puzzle (Level1). In this example, the goal is:"*Complete the block puzzle based on the target graphic.*".

**Introduction**

This task requires the agent to reconstruct an abstract target image by assembling scattered puzzle pieces from its backpack based on the visual reference.

**Goal**

There is a target item shown on the left margin. You need to fill the correct piece(s) from the backpack to complete the missing part(s) of the frame in the scene, ensuring they match and align with the target item.

**Actions**

place the piece from backpack $\langle BAG.ID \rangle$ into the grid at position $\langle GRID.ID \rangle$

**Difficulty Level**

Level1: The agent needs to select $\underline{1}$ block piece to fill in the missing part.
Level2: The agent needs to select $\underline{2}$ block pieces to fill in the missing part.
Level3: The agent needs to select $\underline{3}$ block pieces to fill in the missing part.

**Example** (see Figure 8)

- Step1
    - Action List
        * A) 'place piece in backpack D into the grid at position I'
        * B) 'place piece in backpack C into the grid at position I'
        * C) 'place piece in backpack A into the grid at position I'
        * D) 'place piece in backpack B into the grid at position I'

### F.6 FILLING (FI)

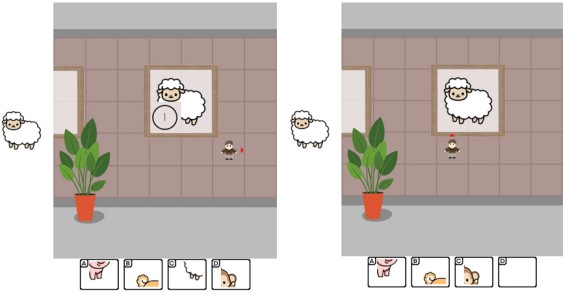

Figure 9: An example of task Filling (Level1). In this example, the goal is: "*Complete the animal puzzle based on the target graphic.*".

**Introduction**

This task requires the agent to reconstruct a figurative animal image by assembling scattered pieces from its backpack based on the visual reference.

**Goal**

There is a target item shown on the left margin. You need to fill the correct piece(s) from the backpack to complete the missing part(s) of the frame in the scene, ensuring they match and align with the target item.

**Actions**

place the piece from backpack $\langle BAG.ID \rangle$ into the grid at position $\langle GRID.ID \rangle$

**Difficulty Level**

Level1: The agent needs to select $\underline{1}$ piece to fill in the missing animal part.
Level2: The agent needs to select $\underline{2}$ pieces to fill in the missing animal part.
Level3: The agent needs to select $\underline{3}$ pieces to fill in the missing animal part.

**Example** (see Figure 9)

- Step1
    - Action List
        * A) 'place the piece from backpack D into the grid at position I'
        * B) 'place the piece from backpack B into the grid at position I'
        * C) 'place the piece from backpack C into the grid at position I'
        * D) 'place the piece from backpack A into the grid at position I'

### F.7 MEMORY FILLING (MFI)

**Introduction**

This task requires the agent to remember a figurative animal in the hint bar and reconstruct it by assembling scattered pieces from its backpack.

**Goal**

In the first image, a target item will be shown on the left margin that you need to remember. In the following images, the target item will not be shown. You need to fill the correct piece(s) from the backpack to complete the missing part(s) of the frame in the scene, ensuring they match and align with the target item. If you understand the rules, choose 'continue' to begin the task.

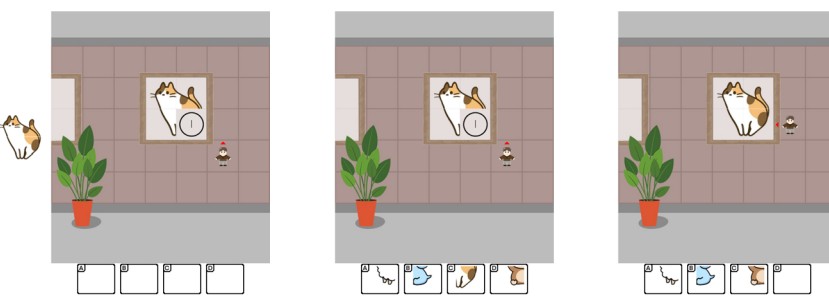

Figure 10: An example of task Memory Filling (Level1). In this example, the goal is: "*Remember and complete the animal puzzle based on the target graphic.*".

**Actions**

place piece from backpack $\langle BAG.ID \rangle$ into the grid at position $\langle GRID.ID \rangle$

**Difficulty Level**

Level1: The agent needs to remember and select **1** piece to fill in the missing animal part.
Level2: The agent needs to remember and select **2** pieces to fill in the missing animal part.
Level3: The agent needs to remember and select **3** pieces to fill in the missing animal part.

**Example** (see Figure 10)

- Step1

  – Action List

    * A) 'continue'

- Step2

  – Action List

    * A) 'place piece from backpack D into the grid at position I'
    * B) 'place piece from backpack A into the grid at position I'
    * C) 'place piece from backpack C into the grid at position I'
    * D) 'place piece from backpack B into the grid at position I'

### F.8    MAZE (MA)

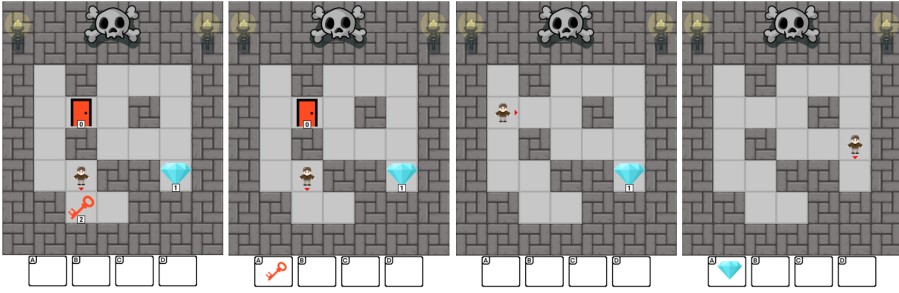

Figure 11: An example of task Maze (Level1). In this example, the goal is: "*Use the orange key to open the orange door and then obtain the diamond.*".

**Introduction**

This task requires the agent to use the keys to unlock corresponding doors to get the diamond.

**Goal**

There is a diamond shown in the scene, and you need to obtain the diamond. When your path is blocked by a door, you can use a key of the same color to unlock it. Note: You must pick up the key first before you can use it to unlock doors.

**Actions**

obtain item with label $\langle ITEM.ID \rangle$
use the key in backpack $\langle KEY.ID \rangle$ to unlock door with label $\langle DOOR.ID \rangle$

**Difficulty Level**

Level1: The agent needs to open **1** door to get to the diamond.
Level2: The agent needs to open **2** doors to get to the diamond.
Level3: The agent needs to open **3** doors to get to the diamond.

**Example** (see Figure 11)

- Step1

    - Action List

        * A) 'obtain item with label 1'
        * B) 'obtain item with label 2'

- Step2

    - Action List

        * A) 'use the key in backpack A to unlock door with label 0'
        * B) 'obtain item with label 1'

- Step3

    - Action List

        * A) 'obtain item with label 1'

## F.9 DECODE MAZE (DMA)

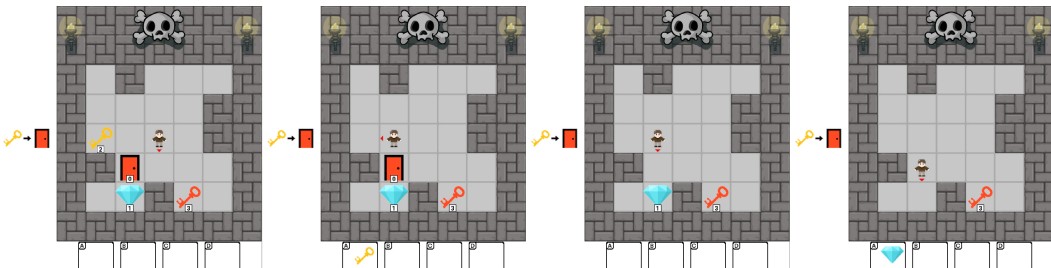

Figure 12: An example of task Decode Maze (Level1). In this example, the goal is: "*Use the yellow key to unlock the red door, then obtain the diamond.*".

**Introduction**

This task requires the agent to leverage the hint information to make correct choices and formulate a series of plans to obtain the diamond as few steps as possible.

**Goal**

There is a diamond in the scene, and your goal is to obtain it. Some paths are blocked by doors, and the key required to unlock each door color is shown in the left hint panel. You must consult the hint panel and use the specified key to open the corresponding door.

**Actions**

obtain item with label $\langle ITEM.ID \rangle$
use the key in backpack $\langle KEY.ID \rangle$ to unlock door with label $\langle DOOR.ID \rangle$

**Difficulty Level**

Level1: The agent needs to open **1** door to get to the diamond.
Level2: The agent needs to open **2** doors to get to the diamond.
Level3: The agent needs to open **3** doors to get to the diamond.

**Example** (see Figure 12)

- Step1

    – Action List

        * A) 'obtain item with label 3'
        * B) 'obtain item with label 2'
        * C) 'obtain item with label 1'

- Step3

    – Action List

        * A) 'use the key in backpack A to unlock door with label 0'
        * B) 'obtain item with label 3'
        * C) 'obtain item with label 1'

- Step4

    – Action List

        * A) 'obtain item with label 1'
        * B) 'obtain item with label 3'

### F.10 MEMORY MAZE (MMA)

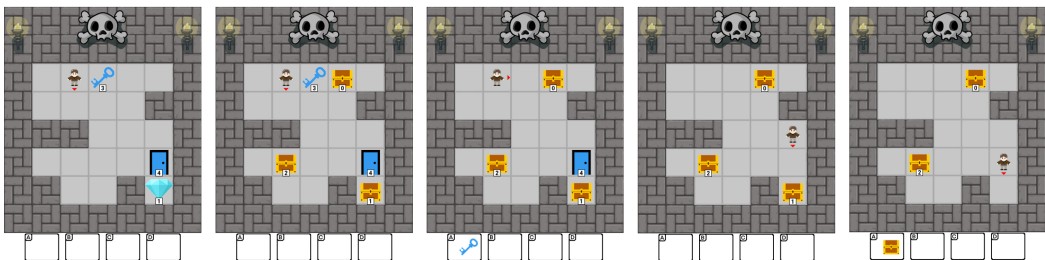

Figure 13: An example of task Memory Maze (Level1). In this example, the goal is: *"Use the blue key to unlock the blue door, then obtain the diamond hidden in the treasure chest using your memory."*.

**Introduction**

This task requires the agent to remember the location of the diamond and use the keys to unlock corresponding doors to get the diamond.

**Goal**

In the first image, a diamond will be shown in the scene that you need to remember its location. In the following images, the diamond will not be shown and several treasure boxes will be generated in the scene. You must choose to open the treasure box located at the diamond's original position to obtain the diamond. When your path is blocked by a door, you can use a key of the same color to unlock it. Note: You must obtain the key before you can use it to unlock doors. If you understand the rules, choose 'continue' to begin the task.

**Actions**

obtain item with label $\langle ITEM.ID \rangle$
use the key in backpack $\langle KEY.ID \rangle$ to unlock door with label $\langle DOOR.ID \rangle$

**Difficulty Level**

Level1: The agent needs to open **1** door to get to the diamond.
Level2: The agent needs to open **2** doors to get to the diamond.
Level3: The agent needs to open **3** doors to get to the diamond.

**Example** (see Figure 13)

- Step1
  - Action List
    * A) 'continue'
- Step2
  - Action List
    * A) 'obtain item with label 0'
    * B) 'obtain item with label 1'
    * C) 'obtain item with label 3'
    * D) 'obtain item with label 2'
- Step3
  - Action List
    * A) 'use the key in backpack A to unlock door with label 4'
    * B) 'obtain item with label 2'
    * C) 'obtain item with label 1'
    * D) 'obtain item with label 0'
- Step4

    – Action List

        ∗ A) 'obtain object with label 1'

        ∗ B) 'obtain item with label 0'

        ∗ C) 'obtain item with label 2'

## F.11 SORTING (SO)

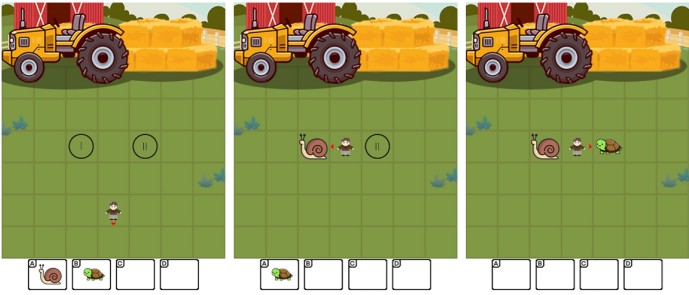

Figure 14: An example of task Sorting (Level1). In this example, the rule is: "*The lighter the animal is, the faster it is.*" and the goal is: "*Rank the animal in the backpack from fast to slow by speed in position I, II*".

**Introduction**

This task requires the agent to sort items based on a provided rule, even if the rule contradicts real-world knowledge.

**Goal**

$\langle RULE \rangle$. Rank the $\langle TYPE \rangle$ in the backpack by $\langle PROPERTY \rangle$ in position I, II.

**Actions**

place $\langle TYPE \rangle$ from backpack $\langle BAG.ID \rangle$ into the grid at position $\langle GRID.ID \rangle$

**Difficulty Level**

Level1: The agent needs to learn the new rule and sort **2** animals in corresponding order.
Level2: The agent needs to learn the new rule and sort **3** animals in corresponding order.
Level3: The agent needs to learn the new rule and sort **4** animals in corresponding order.

**Example** (see Figure 14)

- Step1

    – Action List

        ∗ A) 'place animal from backpack B at grid I'

        ∗ B) 'place animal from backpack A at grid II'

        ∗ C) 'place animal from backpack A at grid I'

        ∗ D) 'place animal from backpack B at grid II'

- Step2

    – Action List

        ∗ A) 'place animal from backpack A at grid II'

### F.12 Placement (PL)

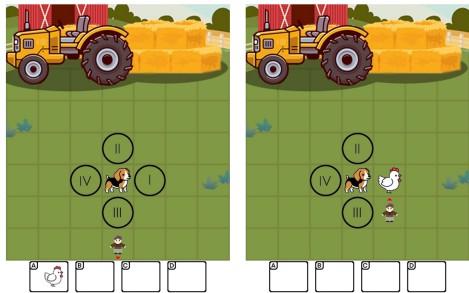

Figure 15: An example of task Placement (Level1). In this example, the goal is: "*Place the chick in the opposite direction to the west of the dog.*".

**Introduction**

The task requires the agent to place the item in the specified location.

**Goal**

A direction will be provided: $\langle ORIENTATION \rangle$. Determine its opposite direction, and then place $\langle ITEM_1 \rangle$ in the corresponding location around $\langle ITEM_2 \rangle$.

**Actions**

place $\langle ITEM \rangle$ into the grid at position $\langle GRID.ID \rangle$

**Difficulty Level**

Level1: The agent needs to place the item in opposite positions of the given position in **4** places.
Level2: The agent needs to place the item in opposite positions of the given position in **8** places.
Level3: The agent needs to place the item in opposite positions of the given position in **8** places, and then **turn one grid clockwise or counterclockwise**.

**Example** (see Figure 15)

- Step1

    - Action List

        * A) 'place elephant into the grid at position III'
        * B) 'place elephant into the grid at position I'
        * C) 'place elephant into the grid at position II'
        * D) 'place elephant into the grid at position IV'

## G In-Content Learning Examples

**Example of Classification Task**

**Step1**: Since one of the goals is to 'place sushi in the green basket', and the item with label 2 is a sushi, the first action you should choose is 'pick up the item with label 2'.
**Step2**: After picking up the sushi, you need to place it in the green basket (label 3). So, the next action should be 'put the item from backpack A into the basket with label 3'.
**Step3**: You have already completed the first goal. The next goal is to 'place pizza in the red basket'. Therefore, the correct action is 'pick up the item with label 1'.
**Step4**: Finally, you need to place the pizza in the red basket (label 0) after picking it up. So, the last action should be 'put the item from backpack A into the basket with label 0'.

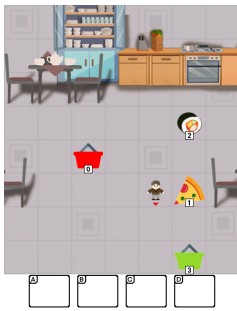

Figure 16: "*Example image for Classification Task*"

**Example of Counting Task**

**Step1**: Since the goal is to 'collect 1 egg', and the item with label 1 is 1 egg. So the first action you should choose is 'pick up egg with label 1'.
**Step2**: After picking up egg with label 1, you have already reached the goal. So the next action you should choose is 'I have already collected 1 eggs'.

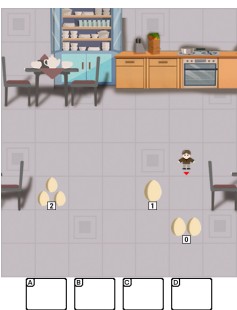

Figure 17: "*Example image for Counting Task*"

**Example of Selection Task**

**Step1**: You should remember the item shown on the left margin first. The item is a sushi. After remembering it, the first action you should choose is 'continue'.
**Step2**: After you remembering it, you see that several items are displayed in the frame. Since you remember the target item is a sushi and the item with label 0 is a sushi, the next action you should choose is 'choose food with label 0'.

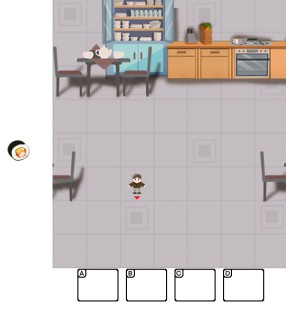

Figure 18: "*Example image for Selection Task*"

**Example of Memory Decode Task**

**Step1**: Since the correspondence(s) will be shown only in the firat image, you should remember the correspondence first. The correspondence is: 'hamburger corresponds to football'. Also the goal say: 'If you understand the rules, choose 'continue' to begin the task.' After remembering it, the first action you should choose is 'continue'.

**Step2**: Since the final goal is to 'select the correct corresponding item to the target item inside the black box in the upper left corner', you have already remembered the correspondence: 'hamburger corresponds to football', the target item in the top left corner is hamburger, so you need to select the football on the right part of the frame. Since the item with label 2 is the football, the first action you should choose is 'choose item with label 2'.

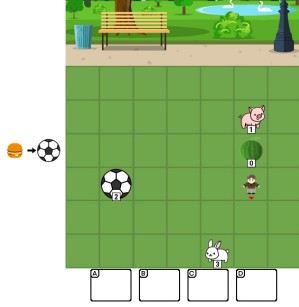

Figure 19: "*Example image for Memory Decode Task*"

**Example of Puzzle Task**

**Step1**: Since the goal is to 'complete the missing part(s) of the frame in the scene', you need to analyze the information of the missing part of the picture according to the information of the known picture. The framed image on the right is missing the upper-left corner, specifically a left triangle with a right Angle vertex in the upper right corner. By examining the four available puzzle pieces, we can determine that piece C matches the missing feet. So the action you should choose is 'place piece in backpack C at grid I'.

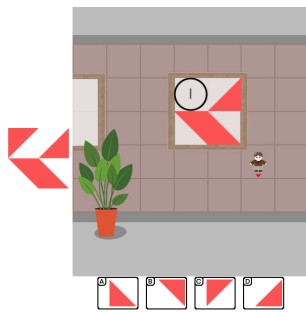

Figure 20: "*Example image for Puzzle Task*"

**Example of Filling Task**

**Step1**: Since the goal is to 'complete the missing part(s) of the picture frame in the scene', you need to analyze the information of the missing part of the picture according to the information of the known picture. The item on the left shows a pink pig, while the framed image on the right is missing the lower-right corner, specifically the front feet of the pig. By examining the four available puzzle pieces, it's sure that piece C matches the missing feet. In addition to shape, the color of the piece further confirms that piece C is correct, as its color matches that of the target image, unlike

the other options. So the action you should choose is 'place piece in backpack C at grid I'.

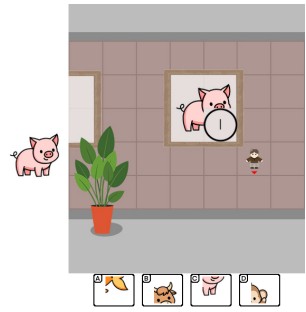

Figure 21: "*Example image for Filling Task*"

**Example of Memory Filling Task**

**Step1**: Since the target item will be shown only in the firat image, you should remember the diagram first. The diagram is a pink pig. After remembering it, the first action you should choose is 'continue'.

**Step2**: Since the final goal is to 'complete the missing part(s) of the picture frame', you need to analyze the information of the missing part of the picture according to the information of the known picture. You have already remembered the target image on the left shows a pink pig, while the framed image on the right is missing the lower-right corner, specifically the front feet of the pig. By examining the four available puzzle pieces, it's sure that piece C matches the missing feet. In addition to shape, the color of the piece further confirms that piece C is correct, as its color matches that of the target image, unlike the other options. So the first action you should choose is 'place piece in backpack C at grid I'.

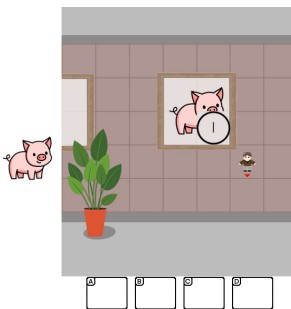

Figure 22: "*Example image for Memory Filling Task*"

**Example of Maze Task**

**Step1**: Since one of the goals is to 'obtain the diamond', and your path is blocked by a blue door, so you should first pick up the blue key. Since the object with label 0 is blue key, the first action you should choose is 'obtain object with label 0'.

**Step2**: After ficking up blue key, you should use the blue key to open the blue blue door. The blue key is in backpack A and the door with label 2 is the blue door, the next action you should choose is 'use the key in backpack A to unlock door with label 2'.

**Step3**: After that, you can get the diamond directly. The object with label 1 is the dimand, so in the next step, we choose the option: 'obtain object with label 1'.

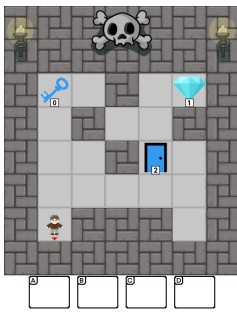

Figure 23: "*Example image for Maze Task*"

**Example of Decode Maze Task**

**Step1**: The path to obtaining the diamond is blocked by the red door. And from the left hint panel, it can be seen that a yellow key is needed to open the red door. The label of the yellow key is 2, so the first action you should choose is C: 'obtain item with label 2'.
**Step2**: After remembering the diamond's label, you can see that your path is blocked by a red door, so you should first pick up the blue key. Since the object with label 1 is blue key, the first action you should choose is B: 'obtain object with label 1'.
**Step3**: After that, you can get the diamond directly. Since the item with label 1 is the dimand, the next action you should choose is A: 'obtain item with label 1'.

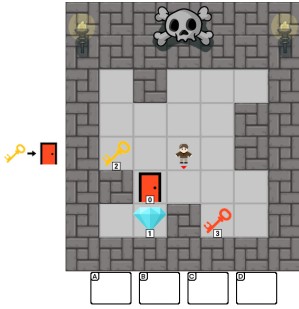

Figure 24: "*Example image for Decode Maze Task*"

**Example of Memory Maze Task**

**Step1**: Since the goal is to 'memorize the exact location of the diamond and obtain the diamond in the dungeon', you should remember the diamond's position and label first. The diamond is of label 3. After remembering it, the first action you should choose is 'continue'.
**Step2**: After remembering the diamond's label, you can see that your path is blocked by a blue door, so you should first pick up the blue key. Since the object with label 1 is blue key, the next action you should choose is 'obtain object with label 1'.
**Step3**: After ficking up blue key, you should use the blue key to open the blue blue door. The blue key is in backpack A and the door with label 2 is the blue door, the next action you should choose is 'use the key in backpack A to unlock door with label 2'.
**Step4**: After that, you can get the diamond directly. Since you have remembered in the first step that the object with label 3 is the dimand, the next action you should choose is 'obtain object with label 3'.

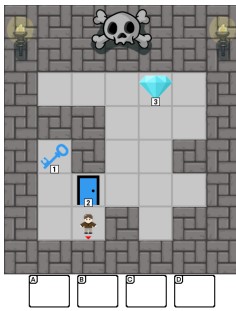

Figure 25: "*Example image for Memory Maze Task*"

**Example of Sorting Task**

**Step1**: The new rule is 'the lighter the animal is, the slower it is' and the goal is to 'rank the animal in the backpack from slow to fast by speed in position I, II', the animal in backpack A is a mouse and in backpack B is an elephant. Since the mouse is lighter than the elephant, so it is slower. Since we should rank the animals from slow to fast in grid I and II, the mouse should be placed at grid I. The first action you should choose is 'place animal from backpack A at grid I'.
**Step2**: Since the elephant is heavier, it is faster. So the elephant should be placed at grid II. Now the elephant is in backpack A, the next action you should choose is A: 'place animal from backpack A at grid II'.

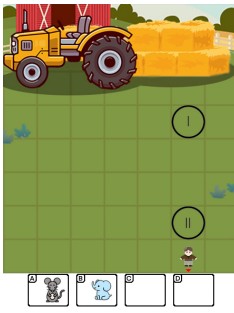

Figure 26: "*Example image for Sorting Task*"

**Example of Placement Task**

**Step1**: Since we need to 'determine its opposite direction', and the direction provided is east, the opposite direction of east is west. So you should place the cherry on the west direction of the hamburger, which is grid III. So the action you should choose is 'place cherry at grid III'.

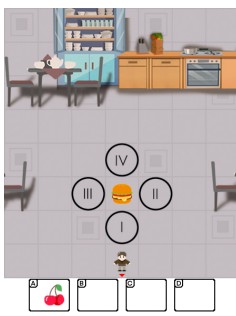

Figure 27: "*Example image for Placement Task*"