# OpenReview forum: "Children's Intelligence Tests Pose Challenges for MLLMs? KidGym: A 2D Grid-Based Reasoning Benchmark for MLLMs"
_ICLR.cc/2026/Conference — ICLR 2026 Poster_

### Official Review · Reviewer_fF2z · 2025-10-23

**Soundness:** 3
**Presentation:** 2
**Contribution:** 3
**Rating:** 6
**Confidence:** 3

**Summary:**

The paper introduces KidGym, a benchmark that assesses five essential capabilities of MLLMs: execution, perception reasoning, learning, memory, and planning. The benchmark is cognitively inspired and uses games as benchmarking tools. The benchmark identifies key limitations of current MLLMs.

**Strengths:**

1. The benchmark captures current limitations of MLLMs: namely, in reasoning over nonsemantic, abstract visual information; insensitivity to item quantity; composite capacity tasks involving the interaction of multiple rules; and weak performance in perception reasoning and planning tasks.
2. The tasks are well designed and include extensive ablations of the environment that allow for more robust evaluation of capabilities. The authors have also made the benchmark customizable to accommodate more tasks, which is a good contribution for future evaluations. The benchmark also varies the level of difficulty for each task type in a nicely controlled way.
3. The authors provide human baselines to compare with MLLLM performance.
4. The authors also tested CoT and ICL methods as additional ablations. It is interesting that CoT improves performance for some models.

**Weaknesses:**

1. Minor: Figure one shows the task types in KidGym. It would be helpful to also illustrate how each task type maps to the four capabilities that KidGym assesses
2. It would be helpful to provide more context for the Wechsler Intelligence Scale. It is unclear where the tasks come from and how they are constructed.
3. The amount of tasks (despite the ablations of environment and randomness) is quite small compared to other existing benchmarks
4. The skills required for each task are largely based on intuitions. The mapping from task to skills would benefit from clearer explanations and justifications based on existing literature.
5. The paper would benefit from more comprehensive error analysis of specific failure cases and whether any patterns can be surfaced from these cases (and providing concrete examples would help the reader get a better sense of the failure modes).

**Questions:**

1. In the abstract, the authors write that the progression “parallels the developmental trajectory from language acquisition to visual perception in children”. Is this true? Does visual perception occur after language acquisition, or do they develop concurrently?
2. Table 1 is very helpful for illustrating the contributions of KidGym. However, the dimensions “Difficulty Level” and “Use Extensible” seem more trivial than aspects like “dynamic vs. static”, which seems to be the more important contribution of KidGym. I would suggest revisiting this table to better show what makes KidGym an important contribution and what distinguishes it from existing benchmarks for MLLMs.
3. Based on the task description, the Sorting task sounds like a linguistic task. Why is it particularly helpful for evaluating MLLMs?
4. Would be helpful to add the human baseline result to Table 3
5. Table 3 can benefit from a clearer caption. For example, what does L stand for in the header? What’s 1, 2, and 3? What does the scale represent? Is the number accuracy?
6. The observation that closed source models outperform open source model is less interesting than comparing general model capabilities, model size, and other factors. Why do you think these models perform better?

---

> ### Author Response · Authors · 2025-11-21
> **W1 to Reviewer fF2z**
>
> Thank you for recognizing the significance of our work. We sincerely appreciate your careful feedback and have provided detailed responses to your questions and concerns below:
>
> **W1 (Figure 1 - Add Task Type)**
>
> Following your suggestion, we have revised Figure 1 to explicitly illustrate how each task type maps to the five capabilities that KidGym assesses. We believe this change makes the figure more informative and helps readers better understand the relationship between task design and the assessed capabilities.

---

> ### Author Response · Authors · 2025-11-21
> **W2 and W4 to Reviewer fF2z**
>
> **W2 & W4 (Wechsler Connection)**
>
> Please see "Global Response -- The Connection between KidGym and Wechsler" for details.

---

> ### Author Response · Authors · 2025-11-21
> **W3 to Reviewer fF2z**
>
> **W3 (Amount of Tasks)**
>
> We would like to clarify two separate notions of **“amount of tasks”**:
>
> - the number of task families / evaluation dimensions, and
> - the number of concrete instances (episodes / questions) per task family.
>
> | Benchmark                | Task families                                       | Instances |
> | ------------------------ | --------------------------------------------------- | --------- |
> | MME-Reasoning            | 3 reasoning types decomposed into 6 capabilities    | 1,188     |
> | MaRs-VQA (COLM 2025)     | 1 (matrix-reasoning problems)                       | 1,440     |
> | MV-MATH (CVPR 2025)      | 11 subject areas × 3 difficulty levels              | 2,009     |
> | MMMU (CVPR 2024)         | 6 major fields spanning 30 subjects                 | 11,500    |
> | SEED-Bench-1 (CVPR 2024) | 12 evaluation dimensions                            | 19,000    |
> | SEED-Bench-2             | 27 evaluation dimensions                            | 24,000    |
> | KidGym                   | 5 capabilities, 12 task types × 3 difficulty levels | Unlimited |
>
> As Table shows, the number of task families in KidGym is on the same order as recent MLLM reasoning benchmarks. Thus, KidGym is not “small” in terms of the diversity of task families, it is in line with the typical granularity of current reasoning-oriented benchmarks.
>
> By contrast, the listed benchmarks are built from **static** **QA** **items** (typically image + question → answer), so their number of instances is fixed once the dataset is constructed. KidGym instead provides an **interactive, dynamic environment**: each task type is instantiated as a multi-step episode in a 2D grid world with randomized layouts, object placements, and initial states. This procedural generation makes the space of problem instances effectively unbounded — users can sample arbitrarily many episodes per task type, far beyond the finite number we use in our experiments for computational reasons. Practically, this is implemented via a Gym interface: a single `env.reset()` call yields a new randomized episode, making it easy to generate large-scale evaluation or training sets without additional manual annotation.

---

> ### Author Response · Authors · 2025-11-21
> **W5 to Reviewer fF2z**
>
> **W5 (Error Analysis)**
>
> We agree that understanding how models fail is crucial for explaining their behavior. Accordingly, we conducted targeted error analyses, particularly on tasks **where the closed-source models performed poorly**, and we systematically compared the three reasoning paradigms (Zero-shot, CoT, and ICL) for each capability.
>
> First, regarding error types, we observe several recurring patterns in Counting (CO) and Puzzle (PU), which are two of the most challenging (“major failure”) tasks in our benchmark:
>
> - **Quantity Misperception Errors (CO)**: In the CO task, the agent sometimes misinterprets “a group of items” as a single object. For example, three apples placed in one grid cell may be treated as “one” instead of “three,” leading to systematic miscounting.
> - **Language-Based Representation Confusion (PU)**: In the PU task, the agent tends to rely heavily on language to represent abstract shapes and spatial relations (e.g., “upper-right triangle”) when describing blocks. However, because such abstract shapes are inherently difficult to encode precisely in natural language, the resulting descriptions often become ambiguous or internally inconsistent, which in turn causes confusion when matching the description to the candidate options.
>
> Second, regarding the systematic comparison of reasoning paradigms (Zero-shot, CoT, ICL), we summarize their strengths and limitations across capability types as follows:
>
> | Zero-shot/CoT/ICL | CL-L1 (Execution)      | PL-L1 (Execution + Learning) | SO-L1 (Execution + Learning) | SE-L1 (Memory)         |
> | ----------------- | ---------------------- | ---------------------------- | ---------------------------- | ---------------------- |
> | GPT-4o            | 0.46/**0.95**/0.32     | 0.71/**0.95**/0.62           | 0.48/**0.83**/0.46           | **1.00**/**1.00**/0.80 |
> | Gemini-2.5        | 0.83/0.86/**0.99**     | **1.00**/**1.00**/**1.00**   | 0.69/**0.84**/0.72           | **1.00**/**1.00**/0.82 |
> | Claude-3.7        | **0.98**/**0.98**/0.69 | **0.97**/**0.97**/0.61       | 0.85/**0.98**/0.52           | **0.97/0.97**/0.47     |
>
> - **Chain-of-Thought (CoT)**: By comparing the performance of the three execution-focused tasks under Zero-shot and CoT (CL/PL/SO), we find that CoT significantly boosts performance on tasks that emphasize execution capability, whose core requirements are preciese instruction following and accurate action selection. In such settings, step-by-step reasoning encourages the model to explicitly consider the rationale behind each intermediate step, thereby directly reducing execution errors.
> - **In-context Learning (ICL)**: We find that ICL is not uniformly advantageous and can even underperform Zero-shot on certain capability types. In particular, for tasks that emphasize memory (e.g., SE) and learning/adaptation (e.g., SO), models sometimes overfit to the provided examples and underweight the actual attributes of the current scene. In these cases, the model appears to “replicate” patterns from the demonstrations rather than dynamically integrating new environmental cues, which can harm generalization. This suggests that when the task type remains the same but the specific scenes change, ICL may be less effective than a simple Zero-shot strategy.
>
> We will incorporate these new findings into the revised version of the paper, where we will summarize the above error taxonomy and capability-based analysis of reasoning paradigms. This will enable readers to clearly see where each paradigm (Zero-shot, CoT, ICL) succeeds or fails across different task types, as well as the underlying causes of these situations.

---

> ### Author Response · Authors · 2025-11-21
> **Q1 to Reviewer fF2z**
>
> **Q1 (Developmental Trajectory)**
>
> Our intention in the abstract was aimed to draw a **metaphorical contrast** between purely linguistic reasoning and more complex **vision–language integrated reasoning**, suggesting a progression in task complexity in our benchmark, not a literal developmental timeline.
>
> From a developmental perspective, we fully acknowledge that **visual perception emerges very early in infancy**, and that language later interacts with and shapes visual processing[1, 2].
>
> Our current phrasing “from language acquisition to visual perception” is therefore potentially misleading and may be read as a chronological claim, which is not what we intended. We will revise the abstract to remove this temporal implication and rephrase the sentence to emphasize the contrast between **language-only reasoning** and **multimodal (vision–language) reasoning**, without suggesting that visual perception follows language acquisition in child development.
>
> [1] Xu, F., & Spelke, E. S. (2000). Large number discrimination in 6-month-old infants. Cognition, 74(1), B1-B11.
>
> [2] LoBue, V., & Rakison, D. H. (2013). What we fear most: A developmental advantage for threat-relevant stimuli. Developmental Review, 33(4), 285-303.

---

> ### Author Response · Authors · 2025-11-21
> **Q2 to Reviewer fF2z**
>
> **Q2 (Table 1 - Add Important Contribution)**
>
> We appreciate the reviewer’s helpful observation regarding Table 1. In the revised version, we have updated Table 1 to better highlight the core contributions of KidGym — namely dynamic vs. static environments, multi-dimensional assessment of cognitive abilities, extensibility and difficulty level. We believe the revised table more clearly conveys what distinguishes KidGym from existing MLLM benchmarks.

---

> ### Author Response · Authors · 2025-11-21
> **Q3 to Reviewer fF2z**
>
> **Q3 (Linguistic Priority)**
>
> Since KidGym is built for multimodal language models, we adopt the following design principle: every task requires the agent to integrate visual and textual information, such that a model restricted to a single modality cannot successfully complete any task. Within this multimodal setup, different tasks naturally have different positioning, with some placing greater emphasis on language understanding and others on visual understanding.
>
> Considering the differences between SO and DE tasks that both require **Learning** ability:
>
> We refer to tasks such as **Sorting (SO)** as **"language-heavy"**: the main difficulty lies in understanding and applying textual rules, often overriding the model’s prior knowledge (e.g., “the faster the animal, the heavier it is”). However, success in these tasks still depends on accurate visual parsing: if the model misidentifies an object or its location, the prediction will be wrong even with perfect language understanding.
>
> Conversely, tasks such as **Decode (DE)** are **"vision-heavy"**: they mainly test the model’s ability to perceive relations among visual elements in the image, while the accompanying language instructions are deliberately simple (e.g., “select the corresponding item”). Nevertheless, the text remains essential, because the model must first understand the basic rule before it can apply it correctly to the visual scene.

---

> ### Author Response · Authors · 2025-11-21
> **Q4 and Q5 to Reviewer fF2z**
>
> **Q4 & Q5 (Table 3 - Add Human Baseline and Clear Caption)**
>
> We appreciate the reviewer’s insightful comments on Table 3. In the revised manuscript, we have
>
> (i) Included both human and random baseline results.
>
> (ii) Refined the table caption to clearly explain the attributes of each column, including meaning of “L”, task abbreviation and success rates.
>
> We believe these changes make Table 3 more informative and easier for readers to understand.

---

> ### Author Response · Authors · 2025-11-21
> **Q6 to Reviewer fF2z**
>
> **Q6 (Closed Models Observation)**
>
> |                | Execution | Memory | Learning | Planning | Perception Reasoning |
> | -------------- | --------- | ------ | -------- | -------- | -------------------- |
> | QwenVL2.5(72B) | **26**    | **47** | **54**   | 6        | **16**               |
> | InternVL3(78B) | 19        | 21     | 26       | 6        | 13                   |
> | DeepSeekVL2    | 12        | 13     | 20       | **7**    | 11                   |
>
> Based on our experimental results, among the open-source models we evaluated, QwenVL2.5-72B achieves the best overall performance across the five capability dimensions. However, all three models perform poorly on the planning dimension, while QwenVL2.5 demonstrates relatively strong memory and learning capabilities.
>
> We can attribute these performance differences to specific architectural features. On the one hand, model size is a non-negligible factor.  For example, DeepSeekVL2 adopts a Mixture-of-Experts (MoE) architecture with only 4.5B active parameters during inference, which partially accounts for its weaker performance. On the other hand, given that InternVL3-78B and QwenVL2.5-72B share the same language backbone (Qwen2.5-72B), the performance gap between them highlights that visual-text alignment — rather than language modeling capability alone — is the key bottleneck.
>
> To further explore the influence of size on the performance of the same model, we conducted experimental comparisons of **open-source models (QwenVL2.5) of different sizes**. The results are as follows:
>
> |               | QwenVL2.5-3B   | QwenVL2.5-7B   | QwenVL2.5-32B  | QwenVL2.5-72B  |
> | ------------- | -------------- | -------------- | -------------- | -------------- |
> | CL(L1/L2/L3)  | 0.00/0.00/0.00 | 0.23/0.07/0.01 | 0.64/0.35/0.05 | 0.48/0.29/0.01 |
> | SE(L1/L2/L3)  | 0.26/0.09/0.02 | 0.79/0.31/0.16 | 1.00/0.81/0.52 | 0.98/0.84/0.65 |
> | DE(L1/L2/L3)  | 0.36/0.19/0.17 | 0.36/0.27/0.24 | 0.99/0.98/0.95 | 0.83/0.68/0.66 |
> | MA(L1/L2/L3)  | 0.07/0.02/0.00 | 0.00/0.00/0.07 | 0.91/0.15/0.03 | 0.42/0.18/0.03 |
> | FI(L1/L2/L3)  | 0.25/0.08/0.07 | 0.34/0.16/0.07 | 0.60/0.30/0.09 | 0.41/0.24/0.09 |
> | PU(L1/L2/L3)  | 0.20/0.09/0.05 | 0.24/0.07/0.05 | 0.29/0.04/0.06 | 0.29/0.08/0.04 |
> | MA*(L1/L2/L3) | 0.00/0.00/0.00 | 0.00/0.00/0.00 | 0.15/0.00/0.00 | 0.12/0.00/0.00 |
> | DE*(L1/L2/L3) | 0.28/0.16/0.12 | 0.25/0.20/0.18 | 0.94/0.92/0.94 | 0.96/0.93/0.92 |
> | SO(L1/L2/L3)  | 0.41/0.18/0.07 | 0.42/0.20/0.07 | 0.62/0.44/0.11 | 0.68/0.28/0.09 |
> | FI*(L1/L2/L3) | 0.27/0.14/0.03 | 0.31/0.11/0.05 | 0.61/0.19/0.08 | 0.38/0.18/0.08 |
> | PL(L1/L2/L3)  | 0.26/0.10/0.17 | 0.21/0.10/0.11 | 0.67/0.33/0.20 | 0.62/0.27/0.15 |
> | CO(L1/L2/L3)  | 0.00/0.00/0.00 | 0.00/0.00/0.00 | 0.00/0.02/0.00 | 0.00/0.00/0.00 |
>
> Our results show that **increasing model size improves performance on average**, but the effect is clearly sub‑linear and not strictly monotonic. QwenVL2.5‑32B is substantially stronger than 3B/7B across most tasks, while the jump from 32B to 72B brings only modest gains on a few tasks and sometimes no improvement or even a slight decrease. In this sense, QwenVL2.5 appears to saturate around 32B.
>
> From a capability perspective, scaling from 3B → 7B → 32B consistently improves performance across the five capability dimensions. In particular, Planning and Memory benefit the most from scaling: for example, the success rates of MA‑L1 are 0.07, 0.00, and 0.91 for 3B, 7B, and 32B respectively, and the success rates of SE‑L1 are 0.26, 0.79, and 1.00. These trends indicate that QwenVL2.5 (around 32B) already captured most of the practically useful gains for these higher‑level abilities.
>
> It is indeed an interesting phenomenon that the **72B model underperforms the 32B model on some tasks**. The same pattern can be observed in the official QwenVL2.5 report: on high-difficulty multimodal benchmarks such as MMMU and MathVista, the 32B and 72B models achieve comparable performance, yet on specific tasks like MathVision, LVBench, CharadesSTA, and ScreenSpot, the 32B model is in fact slightly stronger. We agree this behavior is worth further discussion, and we plan to study such output mapping failure modes of larger models more systematically in future work.

---

> > ### Comment · Reviewer_fF2z · 2025-11-21
> >
> > Thank you for the update! The responses have addressed my previous concerns. I have increased my score.

---

> > > ### Author Response · Authors · 2025-11-22
> > >
> > > Thank you very much for your kind words and for taking the time to review our new experimental results. We are grateful for your thoughtful feedback and are pleased to hear that the supplementary data has addressed your concerns. We appreciate your updated score and are encouraged by your positive assessment of our work. If you have any further suggestions or questions, please feel free to share. Thank you once again!

---

### Official Review · Reviewer_iozr · 2025-10-30

**Soundness:** 3
**Presentation:** 2
**Contribution:** 3
**Rating:** 4
**Confidence:** 4

**Summary:**

The paper introduces KidGym, a 2D grid-based benchmark for multimodal LLMs (MLLMs), inspired by the Wechsler intelligence scales. It operationalizes five capabilities: execution, perceptual reasoning, learning, memory, and planning, via 12 tasks (6 single-capability and 6 composite), each with three difficulty levels and randomized layouts. Tasks include Classification, Filling, Puzzle, Selection, Decode, Maze, and composite variants like Maze, Decode, Sorting, Filling, Placement, Counting. The environment presents a scene map, hint bar, and backpack with labeled items and exposes high-level actions to avoid low-level control. The authors evaluate seven SOTA MLLMs (closed- and open-source) across 100 zero-shot rounds per task, plus CoT/ICL variants, and provide capability radar plots. Key findings: closed models outperform open models; tasks involving abstract visual patterns, quantities, and composite rules are consistently harder. The benchmark is positioned as customizable (Gym-based) and claims a public release.

**Strengths:**

1. Clear capability decomposition tied to a widely known cognitive framework (Wechsler), with explicit mapping of tasks to abilities. This makes the evaluation scope transparent and multifaceted.
2. Interactive, dynamic evaluation rather than purely static VQA-style prompts. The environment design (randomized layouts, diverse scenes, identification labels, hint/backpack) is thoughtful and motivated by known pain points (context consistency, low-level action brittleness).
3. Breadth of models and tasks. Seven representative models, three difficulty levels, and 12 tasks give a reasonably comprehensive picture.
4. Empirically grounded insights that echo community experience: weaknesses with non-semantic abstract vision (Puzzle), quantity sensitivity (Counting), and multi-rule composition (composites like Maze) are convincingly demonstrated and discussed.
5. Extensibility claim (Gym API, configurable scenes/difficulty) is attractive for community uptake.

**Weaknesses:**

1. Metric and objective mismatch for “planning.” Maze/MA* are described as requiring “as few steps as possible” (§5.1, p. 6), but the reported results are success rates only (Table 3); there is no path-length/efficiency metric, nor time-to-solve, which weakens the planning claim. Without cost-sensitive metrics (or success under step budgets), planning is only partially measured.
2. High-level actions may reduce ecological difficulty. By design, the agent executes macros like “pick up the basketball” rather than navigating. This meaningfully simplifies the control problem and can blur construct validity for execution and planning. An ablation showing how conclusions change under finer-grained actions would strengthen the case.
3. Interpretation of Wechsler analogy may be overextended. The paper frames results as “children’s intelligence tests pose challenges” (title) and borrows the Wechsler taxonomy, but does not provide normative calibration or validated correspondences to psychometric constructs or score scaling. The framing risks being rhetorical rather than psychometrically grounded.
4. Presentation quality / editing. There are numerous typos and grammatical issues (“capbilities,” “cusomization,” “commodate,” etc.), and a few imprecisions (e.g., “diamond is hidden among several treasure chests” but no precise termination rules).

**Questions:**

Please see the weaknesses. If the authors can address my concerns listed in the weaknesses,I'd like to raise my evaluation.

---

> ### Author Response · Authors · 2025-11-21
> **W1 to Reviewer iozr**
>
> Thank you for acknowledging the quality of our work and we have provided detailed responses to your questions and concerns below:
>
> **W1 (Success Rate)**
>
> In our implementation, for all tasks, success is defined with respect to the **theoretical shortest action sequence** in the underlying environment. For each task, we pre-set the optimal path length L* as the maximum number of steps. During evaluation, an episode is counted as successful only if the agent reaches the goal in exactly L* steps; any trajectory that reaches the goal but exceeds L* steps is counted as a failure.
>
> For example, in an MA_L1(Maze, Level1) instance, the agent may be able to finish the task in an optimal way:
>
> - Pick up the correct color key.
> - Open the corresponding color door.
> - Get the diamond.
>
> ```Python
> register(
>     id = 'MA_L1',
>     entry_point = 'task.maze:Maze',
>     kwargs = {'match_pairs': 1, 'max_steps': 3,
>               'high_level': True, 'task_name': 'Maze_L1'}
> )
> ```
>
> In this case, the maximum number of steps for the task is set to 3 (in `task/config.py`), and completing the task in **exactly 3 steps is counted as success**, whereas completing it in 4 or more steps — even if the agent eventually reaches the diamond — is counted as failure.
>
> Thus, the reported “success rate” in Table 3 is already a **cost-sensitive metric** equivalent to **“success under the optimal step budget”**, and therefore does reflect the agent’s planning efficiency rather than mere reachability. We will clarify this definition in the revised manuscript and replace the ambiguous wording “as few steps as possible” with a more precise description based on the optimal path-length constraint.

---

> ### Author Response · Authors · 2025-11-21
> **W2 to Reviewer iozr**
>
> **W2 (High-Level Actions)**
>
> In KidGym, we intentionally adopt an enumerated high-level action space rather than low-level actions. The goal of this design is to **factor out low-level motor control and path-following**, so that the benchmark can focus more directly on the agent’s cognitive ability. We view this as one of the key design features of KidGym, rather than a limitation.
>
> This choice is also consistent with the practice in human intelligence testing, where **IQ tests are not intended to measure fine motor skills**. Instead, they present problems in a way that minimizes motor demands, so that the measured variance primarily reflects cognitive capacities rather than manual dexterity.
>
> If we were to add low-level actions, then even to complete a conceptually simple step such as “pick up the basketball,” the agent would first need to solve a separate navigation problem: deciding where to move next, when to turn, how to reach the target, and so on. This would **introduce additional navigation demands** that are orthogonal to the capacities we aim to isolate, and would make it harder to interpret whether failures arise from high-level decision making or from low-level movement control. On the other hand, when an agent executes tasks in the future, once a high-level goal has been specified (such as determining where to go or which object to pick up), the subsequent action can be fully **delegated to other tools or specialized algorithms**.
>
> To directly address the reviewer’s suggestion, we have added an ablation that compares high-level and low-level control on the **CL_L1** (Classification, Level 1) task. We evaluate **three strong closed-source models** under both settings and obtain the following success rates on CL_L1:
>
> |                   | CL_L1 (high level) | CL_L1 (low level) |
> | ----------------- | ------------------ | ----------------- |
> | o3                | 1.00               | 0.82              |
> | claude-3.7-sonnet | 0.98               | 0.58              |
> | gemini-2.5-flash  | 0.83               | 0.59              |
>
> All three models show a **notable drop in success rate** when required to perform fine-grained navigation and control. In summary, we agree that high-level actions reduce the ecological difficulty of the control problem, but this is a **deliberate trade-off in favor of construct validity**. Contrary to the concern that high-level actions “can blur construct validity for execution and planning,” our intention is precisely that KidGym should reflect the agent’s ability to choose and sequence appropriate high-level actions under given rules and visual states, rather than its ability to micromanage continuous navigation. Our new ablation further shows that moving to finer-grained actions mainly adds an additional layer of difficulty and potential failure modes.

---

> ### Author Response · Authors · 2025-11-21
> **W3 to Reviewer iozr**
>
> **W3 (Wechsler Connection)**
>
> Please see "Global Response -- The Connection between KidGym and Wechsler" for details.

---

> ### Author Response · Authors · 2025-11-21
> **W4 to Reviewer iozr**
>
> **W4 (Presentation Quality)**
>
> We thank the reviewer for pointing out the typos, grammatical issues, and imprecise descriptions. We have carefully proofread and edited the entire manuscript, correcting all spelling and grammar errors. We have also revised several imprecise descriptions and now explicitly specify the termination conditions and success criteria in the task description, so that the procedure and evaluation rules are fully clear.

---

### Official Review · Reviewer_nziT · 2025-11-04

**Soundness:** 3
**Presentation:** 3
**Contribution:** 1
**Rating:** 4
**Confidence:** 4

**Summary:**

This paper introducedsa 2D MLLM benchmark KidGym, incorporating abilities based on the Wechsler Intelligence Scale. The abilities include execution, perception reasoning, learning, memory, planning.

The KidGym benchmark consists of 12 unique tasks (e.g., Classification, Maze, Puzzle, Counting) , each with three difficulty levels. These tasks are designed to assess the five capabilities either individually (6 tasks) or in combination (6 tasks). The environments feature diverse semantic scenes and randomized layouts to ensure robustness and prevent memorization. The benchmark is built on the Gym API, making it extensible.

The authors evaluates seven state-of-the-art MLLMs on KidGym including three closed-source models (o3, GPT-4o , Gemini-2.5-flash, Claude-3.7-sonnet) and four open-sourced models (DeepSeekVL-2, QwenVL-2.5, and InternVL-3).

Their experiments reveal significant limitations in current models, particularly in: reasoning over non-semantic visual information (e.g., the Puzzle task), identifying the quantity of items (e.g., the Counting task), dealing with composite tasks that require multiple capabilities (e.g., Maze* vs. Maze).

**Strengths:**

The benchmark itself is robust. The inclusion of three difficulty levels (L1, L2, L3) effectively demonstrates performance scaling , and the use of randomized layouts prevents models from succeeding via memorization.

KidGym is designed to be fully user-customizable and extensible, and it is built on the standard Gym API. This, along with its public release, makes it a valuable and practical tool for the MLLM community.

**Weaknesses:**

“Single-capability” claim unsubstantiated. The paper asserts six tasks isolate individual abilities but provides no validation (e.g., construct ablations/lesioning). In practice these tasks are compound: Maze (labeled Planning) also needs Perception; Classification (Execution) also needs Perception + Learning. Low scores may reflect hidden prerequisites, not the target skill.

Wechsler Intelligence Scales inspiration is superficial. Tasks don’t map cleanly to established Wechsler subtests/constructs, and no psychometric evidence (reliability/validity) is given.

Related work gaps. Important recent capability benchmarks (e.g., MaRs-VQA in COLM 2025) are missing.

Limited headroom amid fast model progress. Frontier models such as GPT-5 and Gemini 2.5 Pro already approach saturation on several tasks, risking short shelf-life of your benchmark.

**Questions:**

See weaknesses for more details.

---

> ### Author Response · Authors · 2025-11-21
> **W1 to Reviewer nziT**
>
> Thank you for your detailed and insightful feedback. Below, we provide our responses to your questions and concerns:
>
> **W1 (Single/Compound Capability)**
>
> The reviewer's concern touches on a fundamental issue in cognitive measurement known as the "task impurity problem" [1-4]: **no single task can purely measure one cognitive capacity in isolation**. Any real-world task inevitably engages multiple cognitive processes simultaneously, and this is true whether we are evaluating human children or artificial agents.
>
> Given this context, KidGym follows standard practice in psychometrics by designing tasks in which one or two target abilities are dominant by design.
>
> **1. Our Design Principle for** **Capacity** **Mapping**
>
> Single-Capacity tasks are designed such that:
>
> - Performance is primarily driven by variation along one target capacity dimension.
> - Non-target capabilities are explicitly minimized through mechanisms (e.g., keeping information visible to reduce memory load, or providing direct action options to reduce planning depth).
>
> Composite-Capacity tasks are designed such that:
>
> - Success requires coordination of multiple capacities that cannot be suppressed through mechanisms (e.g., remembering hidden information while performing sequential actions).
>
> This approach mirrors standard practice in psychometric test design.
>
> **2. Why Maze (MA) is only mapped to Planning?**
>
> KidGym defines Perception Reasoning as **“drawing inferences and making decisions from visual inputs”** (Section 3). We agree that MA requires basic perception to recognize keys and doors. However, in MA, the visual demands are intentionally kept simple: keys and doors are rendered with highly distinct colors and shapes, and each cell is clearly separated on the grid. Recognizing a “red key” or “blue door” is a low-complexity visual judgment. This is essentially straightforward visual recognition, rather than high-level visual reasoning.
>
> The core challenge in MA lies in **sequential dependence on actions**. The crucial decisions concern ordering and contingencies, such as:
>
> - Which key should be collected first and which order should doors be opened so that the diamond remains reachable?
> - Each action changes the reachable part of the maze, so the agent must perform a multi-step lookahead and construct a strategy.
>
> In other words, MA uses simple, easily distinguishable visual inputs as prerequisites, while the increase in difficulty is driven by the depth and structure of the action sequence, not by visual recognition itself. In addition, MA imposes a step limit: if an agent fails to complete the task within the specified number of steps, the episode is counted as a failure. For these reasons, we primarily map Maze to the **Planning** dimension rather than to Perception Reasoning.
>
> **3. Why Classification (CL) is only mapped to Execution?**
>
> CL follows the following characteristics when it is designed:
>
> - Perception is low-difficulty: the mapping between items and containers is explicit and visually obvious (e.g., fruits, toys, baskets with different colors).
> - No nontrivial rule learning is required: the agent is not asked to infer or generalize a latent rule; it only needs to remember and implement a finite list of directives.
>
> In CL, the main source of error is **instruction following and action selection**. At each step, the agent must choose the correct high-level action (e.g., “put cherry → yellow basket”) from a set of alternatives. All necessary perceptual information is visible, and the rules are already specified; what matters is faithfully executing the given mapping for every item without mis-binding.
>
> In contrast, the Sorting (SO) task requires not only precise action selection but also understanding and applying specific rules (e.g., “the faster the animal, the heavier it is”) in order to succeed. For these reasons, we classify CL as an "Execution-only" task, and reserve the "Execution + Learning" label for tasks such as SO and PL.
>
> [1] Karr, J. E., Areshenkoff, C. N., Rast, P., Hofer, S. M., Iverson, G. L., & Garcia-Barrera, M. A. (2018). The unity and diversity of executive functions: A systematic review and re-analysis of latent variable studies. Psychological bulletin, 144(11), 1147.
>
> [2] Miyake, A., Friedman, N. P., Emerson, M. J., Witzki, A. H., Howerter, A., & Wager, T. D. (2000). The unity and diversity of executive functions and their contributions to complex “frontal lobe” tasks: A latent variable analysis. Cognitive psychology, 41(1), 49-100.
>
> [2] Friedman, N. P., & Miyake, A. (2017). Unity and diversity of executive functions: Individual differences as a window on cognitive structure. Cortex, 86, 186-204.
>
> [4] Miyake, A., & Friedman, N. P. (2012). The nature and organization of individual differences in executive functions: Four general conclusions. Current directions in psychological science, 21(1), 8-14.

---

> ### Author Response · Authors · 2025-11-21
> **W2 to Reviewer nziT**
>
> **W2 (Wechsler Connection)**
>
> Please see "Global Response -- The Connection between KidGym and Wechsler" for details.

---

> ### Author Response · Authors · 2025-11-21
> **W3 to Reviewer nziT**
>
> **W3 (Related Work)**
>
> We thank the reviewer for this valuable suggestion. We will add MaRs-VQA (COLM 2025) to our related work section and include it in Table 1 for a more complete comparison with recent MLLM reasoning benchmarks.
>
> We note that, like KidGym, MaRs-VQA is also inspired by Wechsler-style intelligence assessments for children. However, there are important distinctions in scope and design: MaRs-VQA primarily focuses on matrix reasoning tasks — **a single problem type** that mainly targets perception reasoning abilities. This represents one specific component of the broader Wechsler assessment framework. In contrast, KidGym provides:
>
> - A diverse set of task families covering multiple cognitive capacities.
> - Interactive, stateful episodes that require multi-step reasoning behavior.
> - A psychometrically inspired, multi-dimensional view of MLLM abilities within a unified, embodied environment.
>
> In this sense, KidGym offers complementary insights to MaRs-VQA by providing a broader assessment framework for real-world agent deployment.

---

> ### Author Response · Authors · 2025-11-21
> **W4 to Reviewer nziT**
>
> **W4 (Frontier Models)**
>
> We thank the reviewer for raising this important point. To directly address the concern, we have added new experiments with two frontier models: **GPT-5** and **Gemini-2.5-Pro,** covering all original tasks from **Level-1 to Level-3**, and we further introduce a substantially more challenging upgraded variant of the Memory-Decode task to probe whether headroom still exists at higher difficulty levels.
>
> **Frontier models do not saturate the benchmark**
>
> |               | gpt5           | gemini-2.5-pro |
> | ------------- | -------------- | -------------- |
> | CL(L1/L2/L3)  | 1.00/0.96/0.92 | 0.99/1.00/1.00 |
> | SE(L1/L2/L3)  | 1.00/1.00/0.99 | 1.00/1.00/1.00 |
> | DE(L1/L2/L3)  | 1.00/1.00/1.00 | 1.00/1.00/1.00 |
> | MA(L1/L2/L3)  | 0.97/0.43/0.11 | 0.95/0.18/0.03 |
> | FI(L1/L2/L3)  | 0.74/0.60/0.41 | 0.81/0.66/0.36 |
> | PU(L1/L2/L3)  | 0.30/0.06/0.01 | 0.19/0.13/0.07 |
> | MA*(L1/L2/L3) | 0.62/0.10/0.01 | 0.66/0.49/0.00 |
> | DE*(L1/L2/L3) | 1.00/1.00/1.00 | 1.00/1.00/1.00 |
> | SO(L1/L2/L3)  | 1.00/0.99/0.94 | 0.99/0.99/0.93 |
> | FI*(L1/L2/L3) | 0.77/0.61/0.40 | 0.81/0.66/0.36 |
> | PL(L1/L2/L3)  | 1.00/1.00/0.88 | 1.00/1.00/0.74 |
> | CO(L1/L2/L3)  | 0.36/0.18/0.16 | 0.72/0.36/0.19 |
>
> While GPT-5 and Gemini-2.5-Pro indeed perform very well on some relatively easy tasks (e.g., CL/DE/DE*), their performance is far from saturated on tasks requiring stronger planning (e.g., MA), perception reasoning (e.g., FI / PU) or composite tasks (e.g., MA*/CO). In several task families, while models achieve high success rates at Level 1, their performance declines sharply at Levels 2 and 3. For instance, the success rate of GPT-5 dropped from 0.97 of MA-L1 to 0.43 of L2, and that of Gemini-2.5-Pro decreased from 0.95 of MA-L1 to 0.18 of L2.
>
> **New upgraded Memory-Decode task**
>
> To further test the ceiling of current models, we designed a more difficult variant on DE* task that increases compositional complexity (see Appendix F). The prompt is as follows:
>
> > “In the first image, you’ll see three pairs of items on the left, each connected by an arrow. These show one-to-one mappings that you need to remember. In the following images, the mappings will be hidden and a target item will appear in the black box in the top-left corner. Starting from this target, apply the mapping **\*twice\*** based on your memory, and then choose the corresponding item in the scene that matches the final result.”
>
> In this variant, the mappings are **interdependent and cyclic** (A→B/B→C/C→A), requiring models to perform **two-step chained transformations** from memory.
>
> |                                 | GPT-5 | Gemini-2.5-Pro |
> | ------------------------------- | ----- | -------------- |
> | **Upgraded Task** | 0.00  | 0.01           |
>
> The results show that neither GPT-5 nor Gemini-2.5-Pro can complete this task. In the new task, an agent must jointly align rule comprehension, relational memory, and the final decision — requiring a two-step chain of interdependent mappings — which even the strongest models struggle to perform reliably.
>
> In conclusion, it can be said that KidGym is **far from saturated** at this stage. In existing tasks, frontier models that perform strongly at Level-1 difficulty experience a sharp drop in success rate as the difficulty increases. As models continue to improve, the newly upgraded memory-decoding task also demonstrates that researchers can readily instantiate more challenging variants (for example, with a longer field of view, more complex visual layouts, or tighter coupling between abilities) to probe the same underlying cognitive capacities at higher difficulty levels. In this sense, relatively simple tasks can be retained as basic diagnostic checks, while new or enhanced tasks can be added to preserve a clear evaluation space along the same capacity axis.

---

### Official Review · Reviewer_dWpT · 2025-11-06

**Soundness:** 3
**Presentation:** 3
**Contribution:** 3
**Rating:** 6
**Confidence:** 3

**Summary:**

This paper proposes a new benchmark, KidGym, aimed at evaluating the cognitive capabilities of Multimodal Large Language Models (MLLMs). The inspiration comes from the Wechsler Intelligence Scale for Children. KidGym includes 12 2D grid-based tasks covering five core capabilities: execution, perception reasoning, memory, learning, and planning (Section 3). These tasks are divided into single-capability and composite-capability types, with three difficulty levels (L1–L3), and each round features randomized environments. KidGym is built on the Gym API and supports full customization. The authors evaluate 7 representative SOTA MLLMs (such as GPT-4o, o3, Gemini, QwenVL, InternVL, etc.) and point out that current models still have significant shortcomings in quantity perception, abstract graphic reasoning, and multi-rule composite tasks.

**Strengths:**

1. The authors align the task design with the Wechsler intelligence testing system and the concept of Executive Function from child cognitive psychology (Appendix A.1, Section 3), providing a structurally grounded framework with human cognitive analogy, in line with the current trend in AI evaluation toward psychological testing methods.

2. KidGym evaluates five core capabilities, with 12 tasks of varying difficulty, and distinguishes between single and composite abilities (Section 5), making it more systematic than existing MLLM benchmarks. For example, Table 1 clearly shows KidGym’s advantages in capability dimensions compared to other benchmarks.

3. The benchmark is built using the Gym API, allowing users to create new task scenarios and rules, making it highly extensible.

4. A systematic comparison was conducted on 7 mainstream MLLMs, including both closed-source models (such as o3, Claude, Gemini, etc.) and open-source models (such as QwenVL, InternVL, etc.), and multiple evaluation paradigms were introduced.

**Weaknesses:**

1. Limited novelty of experimental findings: The core findings of the paper—that MLLMs perform poorly in abstract reasoning, precise counting, and multi-step planning—largely confirm known conclusions from other established benchmarks. For example, the difficulty of abstract visual reasoning has already been well documented in benchmarks like ARC-AGI-2 [1]; similarly, the poor performance in the Counting (CO) task is similar to that found in studies [2][3]. Therefore, although the KidGym framework is novel, its final experimental conclusions do not offer significant new insights into the limitations of MLLMs, but rather reaffirm them within a new task set.

2. Limited diagnostic clarity of the benchmark: The paper distinguishes between “Single Capacity” and “Composite Capacity” tasks but does not provide a rigorous methodology for this classification. In Section 5.1, several tasks labeled as “single capacity” clearly involve multiple cognitive abilities according to their own descriptions, yet the paper offers no explanation as to why they are mapped to only one capacity in Table 2. For example, the Puzzle (PU) task, described as “assembling pieces to reconstruct an image,” can reasonably be considered to involve planning, but it is exclusively mapped to perception reasoning, without any explanation as to why other components are excluded. This missing explanation may make the capacity mapping appear somewhat arbitrary, forcing readers to guess the authors’ assumptions and weakening the benchmark’s claim to clearly isolate and measure specific cognitive abilities.

3. Superficial connection between psychometric concepts and actual evaluation metrics: The paper claims to be inspired by the Wechsler Intelligence Scale (Appendix A.1), but the correspondence between these concepts and the actual implementation is weak. For example, the “Execution” ability is explicitly linked to the Processing Speed Index (PSI) in WPPSI-IV, which measures the speed and accuracy of completing simple tasks (Appendix A.2). This analogy seems more like a narrative framing rather than a strictly applied scientific principle.

Ref：
[1]	Chollet F, Knoop M, Kamradt G, Landers B, Pinkard H. Arc-agi-2: A new challenge for frontier ai reasoning systems. arXiv preprint arXiv:2505.11831. 2025 May 17.
[2]	Li J, Zhang X, Zou H, Guo Y, Xu R, Liu Y, Zhu C, He Y, Cui P. COUNTS: Benchmarking Object Detectors and Multimodal Large Language Models under Distribution Shifts. InProceedings of the Computer Vision and Pattern Recognition Conference 2025 (pp. 9186-9198).
[3]	Amini-Naieni N, Han T, Zisserman A. Countgd: Multi-modal open-world counting. Advances in Neural Information Processing Systems. 2024 Dec 16;37:48810-37.

**Questions:**

Here are some concerns I would like to raise. Thanks.

1. Has there been a deeper analysis of the error types made by the models? Besides success rate, were failed cases (e.g., recognition errors, logical confusion, memory lapses) analyzed? This information is crucial for understanding model behavior. Although Section 6.3 mentions that CoT improves Gemini's performance, is there a systematic analysis of the strengths of the three reasoning paradigms (Zero-shot, CoT, ICL) for each capability or task type?

2. In the tasks, which plays a dominant role: visual or language information? Especially in tasks like Sorting or Decode, is the model's success primarily dependent on language understanding (rules, prompts), or on the parsing of visual images?

3. How much does randomized layout affect evaluation results? In Section 4, it is mentioned that each round has randomized layouts. Has the variance in task results been evaluated to verify the robustness of the benchmark?

4. Does model scale lead to linear performance improvement? Is there any observation that larger models perform better across the five capability dimensions? Is there a saturation point for capability scaling?

---

> ### Author Response · Authors · 2025-11-21
> **W1 to Reviewer dWpT**
>
> Thank you for your thoughtful and thorough feedback. We truly appreciate your recognition of the importance of our work and have outlined detailed responses to address your questions and concerns below.
>
> **W1 (Experimental Novelty)**
>
> KidGym's novelty lies in the following four interconnected aspects:
>
> **1. A unified, multi-dimensional framework for systematic comparison**
>
> Unlike existing benchmarks that typically focus on a single axis (e.g., ARC-AGI-2 on abstract reasoning, Count and CountGD on counting), KidGym provides a psychometrically inspired, multi-dimensional view of MLLM abilities within a **UNIFIED** environment.  A unified framework is important because it enables:
>
> - Compare different cognitive capacities under the same protocol: All tasks share the same interactive format, scene generation pipeline, and scoring conventions, eliminating confounds introduced by heterogeneous benchmark designs.
>
> - Reveal capacity interactions: Through carefully designed task variants, we systematically demonstrate how performance degrades when multiple abilities must be coordinated simultaneously in composite tasks. This inter-capacity analysis cannot be obtained by simply merging individual benchmarks.
>
> Thus, while some ability weakness supported in our paper are consistent with prior literature, KidGym contributes a structured, capacity-level picture of how these weaknesses compound each other within a unified cognitive framework.
>
> **2. A rigorous and original design that is less prone to data leakage**
>
> An increasing number of studies have shown that many public benchmarks are constructed from existing data, which may raise concerns about data leakage and potential overfitting of test sets [1]. Consequently, while existing benchmarks report similar conclusions, their results may not always provide a fully robust assessment of model generalization under carefully controlled evaluation settings.
>
> KidGym addresses these issues by defining original task families with programmatic instance generators, randomized layouts, and multiple difficulty levels. This design supports systematic scaling and extension with new variants that probe the same underlying cognitive abilities. Because all scenes are custom-designed and can be regenerated with different random seeds, KidGym is far less prone to data leakage. Agents must genuinely master the cognitive skills required by the tasks to generalize across new instances, enabling a more rigorous evaluation.
>
> **3. Dynamic, Interactive Evaluation in a Gym-Style Environment**
>
> Methodologically, KidGym is implemented as a fully interactive, gym-style environment rather than a static QA dataset. This design evaluates not only whether a model can produce a single correct answer, but also whether it can maintain coherent behaviour across an entire episode, adapt to intermediate feedback, and update its decisions dynamically.
>
> As a result, when a conclusion superficially resembles an existing finding, KidGym strengthens the evidence by demonstrating that the same limitations persist in dynamic, interactive settings with novel instances, rather than only on fixed test sets. This aligns with recent work advocating a shift from static benchmarks toward dynamic evaluation protocols that mitigate data contamination and better probe genuine reasoning and generalization [2]. In this sense, KidGym complements prior static benchmarks by showing that the observed weaknesses of current models are robust to changes in format (from single-turn QA to multi-step interaction) and to continual generation of new task instances.
>
> **4. An Extensible Platform for Long-Term Research**
>
> Beyond being a one-off benchmark, KidGym is designed as an extensible research platform. Researchers can define new tasks, scenes, and difficulty levels tailored to different capability ranges or specific research questions. Given that most open-source models currently perform poorly on KidGym, the framework can be used to generate targeted training data to improve those weaknesses, while for stronger proprietary models it enables the construction of more challenging tasks to probe their upper performance limits. Because the framework is open and modular, the community can progressively enrich the task space while preserving comparability along the same underlying cognitive dimensions.
>
> [1] White, C., Dooley, S., Roberts, M., Pal, A., Feuer, B., Jain, S., … Goldblum, M. (2025). LiveBench: A Challenging, Contamination-Limited LLM Benchmark. In Proceedings of the 13th International Conference on Learning Representations (ICLR 2025).
>
> [2] Fan, L., Hua, W., Li, L., Ling, H., & Zhang, Y. (2024, August). Nphardeval: Dynamic benchmark on reasoning ability of large language models via complexity classes. In Proceedings of the 62nd Annual Meeting of the Association for Computational Linguistics (Volume 1: Long Papers) (pp. 4092-4114).

---

> ### Author Response · Authors · 2025-11-21
> **W2 to Reviewer dWpT**
>
> **W2 (Diagnostic Clarity)**
>
> The reviewer's concern touches on a fundamental issue in cognitive measurement known as the "task impurity problem" [1-4]: **no single task can purely measure one cognitive capacity in isolation**. Any real-world task inevitably engages multiple cognitive processes simultaneously, and this is true whether we are evaluating human children or artificial agents.
>
> Given this context, KidGym follows standard practice in psychometrics by designing tasks in which one or two target abilities are dominant by design.
>
> **1. Our Design Principle for Capacity Mapping**
>
> Single-Capacity tasks are designed such that:
>
> - Performance is primarily driven by variation along one target capacity dimension.
>
> - Non-target capabilities are explicitly minimized through mechanisms (e.g., keeping information visible to reduce memory load, or providing direct action options to reduce planning depth).
>
> Composite-Capacity tasks are designed such that:
>
> - Success requires coordination of multiple capacities that cannot be suppressed through mechanisms (e.g., remembering hidden information while performing sequential actions).
>
> This approach mirrors standard practice in psychometric test design.
>
> **2. Why Puzzle (PU) is Only Mapped to Perception Reasoning**
>
> To clarify this point, we take the PU task proposed by the reviewers as a specific example. The PU task requires the agent to assemble pieces to reconstruct a target image. The reviewer notes that this could involve planning capability. However, we deliberately designed the mechanisms to suppress planning demands:
>
> At Level-1 of PU, the agent only needs to **fill one missing puzzle piece**. This setting is a pure perceptual completion problem and unambiguously targets perception reasoning.
>
> At Level-2/3 of PU, there is **no temporal dependency** between consecutive actions: each placement decision is independent, and one placement does not constrain or determine the correctness of subsequent placements. The agent can place pieces in any order. Moreover, at every step it can inspect both its backpack and the partially assembled puzzle, eliminating the need to check what pieces are available or what has already been placed. This fundamentally differs from planning tasks like Maze (MA), where each step directly affects whether the next step can succeed.
>
> As a result, at each decision point, the agent essentially answers a local perceptual question:
>
> > "Given the currently visible target pattern and the pieces in my backpack, which piece should I place here to match the target?"
>
> This can be solved through **perception reasoning alone** — matching shapes, orientations, and spatial relationships. Importantly, because actions are independent, the agent does not need to perform multi-step lookahead or goal decomposition. Though previous placements were incorrect, the agent can still correctly place the next piece based on local perceptual matching (but this was identified as a failure in our test).
>
> We will make these design principles and their rationale explicit in the revision, strengthening the benchmark's diagnostic clarity and interpretability.
>
> [1] Karr, J. E., Areshenkoff, C. N., Rast, P., Hofer, S. M., Iverson, G. L., & Garcia-Barrera, M. A. (2018). The unity and diversity of executive functions: A systematic review and re-analysis of latent variable studies. Psychological bulletin, 144(11), 1147.
>
> [2] Miyake, A., Friedman, N. P., Emerson, M. J., Witzki, A. H., Howerter, A., & Wager, T. D. (2000). The unity and diversity of executive functions and their contributions to complex “frontal lobe” tasks: A latent variable analysis. Cognitive psychology, 41(1), 49-100.
>
> [3] Friedman, N. P., & Miyake, A. (2017). Unity and diversity of executive functions: Individual differences as a window on cognitive structure. Cortex, 86, 186-204.
>
> [4] Miyake, A., & Friedman, N. P. (2012). The nature and organization of individual differences in executive functions: Four general conclusions. Current directions in psychological science, 21(1), 8-14.

---

> ### Author Response · Authors · 2025-11-21
> **W3 to Reviewer dWpT**
>
> **W3 (Wechsler Connection)**
>
> Please see "Global Response -- The Connection between KidGym and Wechsler" for details.

---

> ### Author Response · Authors · 2025-11-21
> **Q1 to Reviewer dWpT**
>
> **Q1 (Deep Error Analysis)**
>
> We agree that understanding how models fail is crucial for explaining their behavior. Accordingly, we conducted targeted error analyses, particularly on tasks **where the closed-source models performed poorly**, and we systematically compared the three reasoning paradigms (Zero-shot, CoT, and ICL) for each capability.
>
> First, regarding error types, we observe several recurring patterns in Counting (CO) and Puzzle (PU), which are two of the most challenging (“major failure”) tasks in our benchmark:
>
> - **Quantity Misperception Errors (CO)**: In the CO task, the agent sometimes misinterprets “a group of items” as a single object. For example, three apples placed in one grid cell may be treated as “one” instead of “three,” leading to systematic miscounting.
> - **Language-Based Representation Confusion (PU)**: In the PU task, the agent tends to rely heavily on language to represent abstract shapes and spatial relations (e.g., “upper-right triangle”) when describing blocks. However, because such abstract shapes are inherently difficult to encode precisely in natural language, the resulting descriptions often become ambiguous or internally inconsistent, which in turn causes confusion when matching the description to the candidate options.
>
> Secondly, to systematically compare the Zero-shot, CoT, and ICL reasoning paradigms, we selected four representative tasks and summarized their strengths and limitations:
>
> | Zero-shot/CoT/ICL | CL-L1 (Execution)      | PL-L1 (Execution + Learning) | SO-L1 (Execution + Learning) | SE-L1 (Memory)         |
> | ----------------- | ---------------------- | ---------------------------- | ---------------------------- | ---------------------- |
> | GPT-4o            | 0.46/**0.95**/0.32     | 0.71/**0.95**/0.62           | 0.48/**0.83**/0.46           | **1.00**/**1.00**/0.80 |
> | Gemini-2.5        | 0.83/0.86/**0.99**     | **1.00**/**1.00**/**1.00**   | 0.69/**0.84**/0.72           | **1.00**/**1.00**/0.82 |
> | Claude-3.7        | **0.98**/**0.98**/0.69 | **0.97**/**0.97**/0.61       | 0.85/**0.98**/0.52           | **0.97/0.97**/0.47     |
>
> - **Chain-of-Thought (CoT)**: By comparing the performance of the three execution-focused tasks under Zero-shot and CoT (CL/PL/SO), we find that CoT significantly boosts performance on tasks that emphasize execution capability, whose core requirements are preciese instruction following and accurate action selection. In such settings, step-by-step reasoning encourages the model to explicitly consider the rationale behind each intermediate step, thereby directly reducing execution errors.
> - **In-context Learning (ICL)**: We find that ICL is not uniformly advantageous and can even underperform Zero-shot on certain capability types. In particular, for tasks that emphasize memory (e.g., SE) and learning/adaptation (e.g., SO), models sometimes overfit to the provided examples and underweight the actual attributes of the current scene. In these cases, the model appears to “replicate” patterns from the demonstrations rather than dynamically integrating new environmental cues, which can harm generalization. This suggests that when the task type remains the same but the specific scenes change, ICL may be less effective than a simple Zero-shot strategy.
>
> We will incorporate these new findings into the revised version of the paper, where we will summarize the above error taxonomy and capability-based analysis of reasoning paradigms. This will enable readers to clearly see where each paradigm (Zero-shot, CoT, ICL) succeeds or fails across different task types, as well as the underlying causes of these situations.

---

> ### Author Response · Authors · 2025-11-21
> **Q2 to Reviewer dWpT**
>
> **Q2 (Visual or Language Priority)**
>
> Since KidGym is built for multimodal language models, we adopt the following design principle: every task requires the agent to integrate visual and textual information, such that a model restricted to a single modality cannot successfully complete any task. Within this multimodal setup, different tasks naturally have different positioning, with some placing greater emphasis on language understanding and others on visual understanding.
>
> Considering the differences between SO and DE tasks that both require **Learning** ability:
>
> We refer to tasks such as **Sorting (SO)** as **"language-heavy"**: the main difficulty lies in understanding and applying textual rules, often overriding the model’s prior knowledge (e.g., “the faster the animal, the heavier it is”). However, success in these tasks still depends on accurate visual parsing: if the model misidentifies an object or its location, the prediction will be wrong even with perfect language understanding.
>
> Conversely, tasks such as **Decode (DE)** are **"vision-heavy"**: they mainly test the model’s ability to perceive relations among visual elements in the image, while the accompanying language instructions are deliberately simple (e.g., “select the corresponding item”). Nevertheless, the text remains essential, because the model must first understand the basic rule before it can apply it correctly to the visual scene.

---

> ### Author Response · Authors · 2025-11-21
> **Q3 to Reviewer dWpT**
>
> **Q3 (Randomized Layouts)**
>
> To quantify how randomized layouts affect evaluation, we conducted two complementary analyses. For three representative models (o3, gemini-2.5-flash, and claude-3.7-sonnet), we ran two rounds of experiments for each task. In each round, we used five independent evaluation batches, and each batch was assigned a different random seed for layout generation. The first round used our default configuration of 100 episodes (5×100), and the second round used 10 episodes (5×10), simulating a cheaper, low-sample regime.
>
> | STD                              | CL    | SE    | DE    | MA    | FI    | PU    | MA*   | DE*   | SO    | FI*   | PL    | CO    | Avg   |
> | -------------------------------- | ----- | ----- | ----- | ----- | ----- | ----- | ----- | ----- | ----- | ----- | ----- | ----- | --------- |
> | o3 (5*100 Rounds )               | 0.000 | 0.000 | 0.000 | 0.022 | 0.029 | 0.053 | 0.032 | 0.000 | 0.016 | 0.029 | 0.000 | 0.049 | **0.019** |
> | o3 (5*10 Rounds )                | 0.000 | 0.000 | 0.000 | 0.040 | 0.135 | 0.098 | 0.116 | 0.000 | 0.040 | 0.126 | 0.000 | 0.160 | 0.059     |
> | gemini-2.5-flash (5*100 Rounds)  | 0.020 | 0.000 | 0.000 | 0.047 | 0.055 | 0.085 | 0.008 | 0.000 | 0.032 | 0.055 | 0.012 | 0.032 | **0.029** |
> | gemini-2.5-flash (5*10 Rounds)   | 0.089 | 0.000 | 0.000 | 0.102 | 0.163 | 0.074 | 0.116 | 0.000 | 0.178 | 0.132 | 0.040 | 0.135 | 0.085     |
> | claude-3.7-sonnet (5*100 Rounds) | 0.028 | 0.017 | 0.008 | 0.061 | 0.051 | 0.033 | 0.009 | 0.011 | 0.011 | 0.026 | 0.004 | 0.039 | **0.025** |
> | claude-3.7-sonnet (5*10 Rounds)  | 0.116 | 0.400 | 0.040 | 0.120 | 0.135 | 0.102 | 0.000 | 0.074 | 0.074 | 0.147 | 0.000 | 0.185 | 0.116     |
>
>
>
> Under the **5×100** setting, the per-task standard deviations are very small. For example, the average standard deviation across all tasks is 0.019 for o3, 0.029 for gemini-2.5-flash, and 0.025 for claude-3.7-sonnet, with the maximum per-task standard deviation remaining below 0.09 for all three models.
>
> In contrast, under the **5×10** setting, variability increases substantially: the average standard deviation rises to 0.059, 0.085, and 0.116 for the three models respectively, and some tasks exhibit much larger fluctuations (up to 0.4 in the worst case).
>
> These findings highlight the significance of programmatic random layout for reliable evaluation, which is sufficient to support us in conducting extensive tests for similar tasks. Without such randomness, an environment with a small set of manually designed layouts would be highly susceptible to sampling noise, and performance estimates could fluctuate due to a few fixed configurations. By contrast, under our main evaluation setting — with randomized layouts and a sufficient number of episodes per task — success rates are **stable and low-variance across**, indicating that the benchmark provides a robust measure of model performance.

---

> ### Author Response · Authors · 2025-11-21
> **Q4 to Reviewer dWpT**
>
> **Q4 (Model Scale)**
>
> This is a very worthy point to discuss. On the basis of the original, we conducted experimental comparisons of open-source models (QwenVL2.5) of different sizes. The results are as follows:
>
> |               | QwenVL2.5-3B   | QwenVL2.5-7B   | QwenVL2.5-32B  | QwenVL2.5-72B  |
> | ------------- | -------------- | -------------- | -------------- | -------------- |
> | CL(L1/L2/L3)  | 0.00/0.00/0.00 | 0.23/0.07/0.01 | 0.64/0.35/0.05 | 0.48/0.29/0.01 |
> | SE(L1/L2/L3)  | 0.26/0.09/0.02 | 0.79/0.31/0.16 | 1.00/0.81/0.52 | 0.98/0.84/0.65 |
> | DE(L1/L2/L3)  | 0.36/0.19/0.17 | 0.36/0.27/0.24 | 0.99/0.98/0.95 | 0.83/0.68/0.66 |
> | MA(L1/L2/L3)  | 0.07/0.02/0.00 | 0.00/0.00/0.07 | 0.91/0.15/0.03 | 0.42/0.18/0.03 |
> | FI(L1/L2/L3)  | 0.25/0.08/0.07 | 0.34/0.16/0.07 | 0.60/0.30/0.09 | 0.41/0.24/0.09 |
> | PU(L1/L2/L3)  | 0.20/0.09/0.05 | 0.24/0.07/0.05 | 0.29/0.04/0.06 | 0.29/0.08/0.04 |
> | MA*(L1/L2/L3) | 0.00/0.00/0.00 | 0.00/0.00/0.00 | 0.15/0.00/0.00 | 0.12/0.00/0.00 |
> | DE*(L1/L2/L3) | 0.28/0.16/0.12 | 0.25/0.20/0.18 | 0.94/0.92/0.94 | 0.96/0.93/0.92 |
> | SO(L1/L2/L3)  | 0.41/0.18/0.07 | 0.42/0.20/0.07 | 0.62/0.44/0.11 | 0.68/0.28/0.09 |
> | FI*(L1/L2/L3) | 0.27/0.14/0.03 | 0.31/0.11/0.05 | 0.61/0.19/0.08 | 0.38/0.18/0.08 |
> | PL(L1/L2/L3)  | 0.26/0.10/0.17 | 0.21/0.10/0.11 | 0.67/0.33/0.20 | 0.62/0.27/0.15 |
> | CO(L1/L2/L3)  | 0.00/0.00/0.00 | 0.00/0.00/0.00 | 0.00/0.02/0.00 | 0.00/0.00/0.00 |
>
> Our results show that **increasing model size improves performance on average**, but the effect is clearly sub‑linear and not strictly monotonic. QwenVL2.5‑32B is substantially stronger than 3B/7B across most tasks, while the jump from 32B to 72B brings only modest gains on a few tasks and sometimes no improvement or even a slight decrease. In this sense, QwenVL2.5 appears to saturate around 32B.
>
> From a capability perspective, scaling from 3B → 7B → 32B consistently improves performance across the five capability dimensions. In particular, Planning and Memory benefit the most from scaling: for example, the success rates of MA‑L1 are 0.07, 0.00, and 0.91 for 3B, 7B, and 32B respectively, and the success rates of SE‑L1 are 0.26, 0.79, and 1.00. These trends indicate that QwenVL2.5 (around 32B) already captured most of the practically useful gains for these higher‑level abilities.
>
> It is indeed an interesting phenomenon that the **72B model underperforms the 32B model on some tasks**. The same pattern can be observed in the official QwenVL2.5 report: on high-difficulty multimodal benchmarks such as MMMU and MathVista, the 32B and 72B models achieve comparable performance, yet on specific tasks like MathVision, LVBench, CharadesSTA, and ScreenSpot, the 32B model is in fact slightly stronger. We agree this behavior is worth further discussion, and we plan to study such output mapping failure modes of larger models more systematically in future work.

---

### Author Response · Authors · 2025-11-21
**Global Response -- The Connection between KidGym and Wechsler**

We sincerely thank all reviewers for their thoughtful feedback. In this global response, we would like to address a question jointly raised by four reviewers: **The Connection between KidGym and Wechsler**.

First of all, we would like to clarify: It would be unrealistic to directly apply the Wechsler to the assessment of artificial intelligence agents. Our intention is to use the Wechsler framework as a design prior, a conceptually grounded starting point for decomposing cognitive abilities in a multi-dimensional benchmark.

**Why Wechsler cannot be copied over at the ability and experimental levels?**

Humans and MLLMs differ fundamentally in their embodiment and interaction modalities, so a literal transfer of abilities and subtests is inappropriate.

At the **"ability"** level, for example, the Processing Speed Index (PSI) in Wechsler is intended to capture individual differences in cognitive processing speed and sustained attention under time pressure. In the MLLM setting, however, inference latency is dominated by implementation factors (model size, quantization, batching, hardware accelerators, etc.). Treating wall-clock speed as an analog of PSI would therefore conflate engineering with cognitive ability and would not be scientifically meaningful. Similarly, the Working Memory Index (WMI) in Wechsler is designed to measure the ability to temporarily retain a limited amount of information —— content that may be difficult for humans to keep in mind, but relatively easy to operate on once maintained. In contrast, for MLLMs, previously seen images or tokens are typically stored in context buffers, so retaining the information itself is not the main challenge; instead, the difficulty lies in appropriately integrating and using that information within the current context.

At the **"experimental"** level, many Wechsler subtests rely on embodied, sensorimotor interactions that cannot be reproduced for MLLMs. For instance, certain tasks for young children require pointing to or touching parts of their own body, manipulating physical blocks, or drawing symbols by hand. These modalities (proprioception, handwriting) simply do not suit for current MLLMs.

**How do we actually use the Wechsler framework?**

Given these limitations, we do not simply copy or fully implement the Wechsler test. Instead, we take the Wechsler framework as a design premise, since the Wechsler reflect nearly a century of knowledge about how to operationalize human cognitive abilities into measurable constructs. Although MLLMs are clearly not human, they are increasingly deployed in tasks that require heterogeneous cognitive abilities. By borrowing this conceptual structure, KidGym organizes the capabilities that MLLMs need to complete real-world tasks into interpretable dimensions.

Across the entire benchmark development process, we collaborated with our co-authors, who are **experts in child brain science**, to **formulate a rigorous methodology** that informed the design of the tasks. We first distilled each Wechsler index score into an underlying cognitive ability that is meaningful for MLLMs, and then constructed task families that systematically operationalize these abilities within a unified environment. When a Wechsler subtest could be transferred to an agent setting in an appropriate way, we applied only minimal modifications (e.g., replacing paper-and-pencil responses with discrete actions) while preserving the core cognitive demands. For subtests whose original format is not suitable for MLLMs, we co-designed new tasks with the expert, retaining the intended cognitive requirements while recasting the format to be maximally appropriate for MLLMs. This collaborative and methodologically grounded procedure ensures that each ability dimension is supported by a coherent and well-specified set of cognitive requirements.

**Example: “Execution” and its relation to PSI**

- In Wechsler, PSI tasks require children to perform simple, rule-based operations quickly and accurately. The core demand is to follow explicit instructions reliably under time constraints.

- In KidGym, Execution tasks require MLLMs to accurately follow simple, explicitly specified rules in a structured environment (e.g., correctly categorizing objects). The core demand is instruction following and accurate action selection.

We operationalize this by evaluating success rate within a fixed step budget, which captures whether the model can efficiently complete tasks without unnecessary errors or redundant actions. This preserves the conceptual essence of PSI — "fast and accurate performance on relatively simple, well-defined tasks" — while adapting the measurement approach to the MLLM context.

The discussion above clarifies the connection between KidGym and Wechsler. We hope that this addresses the reviewers' concerns. Additionally, we have included this explanation in Appendix A to assist readers in better understanding the underlying scientific principles.

---

### Author Response · Authors · 2025-12-01
**KidGym Benchmark Overview and Reviewer Response Summary**

Dear AC,

We would like to offer a general summary to introduce our benchmark and the review process.

First, we would like to briefly highlight the key advantages and innovations of our benchmark (for specific details, please refer to the response in W1 for `Reviewer dWpT`):

- **Unified Framework for Systematic Comparison**: KidGym enables multi-dimensional comparisons of MLLM abilities within a unified environment, offering insights into the interactions between capabilities.
- **Rigorous and Original Design**: KidGym’s unique, randomized task design mitigates data leakage and ensures a thorough and generalized evaluation.
- **Dynamic, Interactive Evaluation**: KidGym’s gym-style environment evaluates dynamic, multi-step reasoning and generalization, going beyond static benchmarks.
- **Extensible Platform for Long-Term Research**: KidGym is an open, modular platform, ideal for creating and extending tasks, thus enhancing model evaluation over time.

After reviewing the questions raised by the reviewers, we provide a summary response to two common concerns:

- **The Connection Between KidGym and Wechsler**: Wechsler is a widely recognized intelligence test for children, but it cannot be directly applied to MLLMs due to the fundamental differences in embodiment and interaction modalities between human and artificial intelligence. In collaboration with experts in child brain science, we conceptually adapted the framework to align with the functions of MLLMs while retaining the core cognitive aspects of Wechsler, ensuring the validity of the assessment. For a more detailed explanation, please refer to the "Global Response -- The Connection between KidGym and Wechsler", which was endorsed by `Reviewer fF2z`, who expressed a willingness to increase the score.

- **Design Principle for** **Capacity** **Mapping**: The 'task impurity problem' in cognitive measurement emphasizes that no single task can isolate a single cognitive capacity. In response, KidGym adheres to psychometric principles by designing tasks where one or two target abilities dominate, minimizing non-target processes. This approach ensures that cognitive abilities are measured in a meaningful and focused manner. We have provided more detailed explanations of the corresponding relationships between specific tasks and capabilities in W2 for `Reviewer dWpT` and W1 for `Reviewer nziT`.

Additionally, during the review stage, we incorporated four new ablation and comparative experiments. We believe these experiments effectively address the reviewers' concerns and reinforce the rationale behind KidGym's design:

- **Randomness Experiment** (Q3 for `Reviewer dWpT`): To quantify the impact of randomized layouts on evaluation, we conducted two complementary analyses (5×10 and 5×100). In the 5×100 setting, per-task standard deviations were relatively small. These results emphasize the importance of programmatic random layouts in ensuring reliable evaluations and provide a robust foundation for conducting extensive tests on similar tasks.

- **Comparison of Open-Source Models of Different Sizes** (Q4 for `Reviewer dWpT` & Q6 for `Reviewer fF2z`): Our experimental analysis of QwenVL2.5 models of various sizes shows that increasing model size generally enhances performance. From a capability perspective, scaling from 3B → 7B → 32B consistently improves performance across the five capability dimensions. However, interestingly, the 72B model performed worse than the 32B model on certain tasks, which is consistent with findings in the official QwenVL2.5 report.

- **Frontier Models Performance** (W4 for `Reviewer nziT`): We tested frontier models, GPT-5 and Gemini-2.5-Pro, across all original tasks. While they performed well on simpler tasks, they struggled with those requiring advanced planning, reasoning, and composite abilities. Additionally, both models failed the more challenging variant of the Memory-Decode task. This underscores that KidGym is far from saturated and suggests opportunities for introducing more complex KidGym tasks in the future.

- **Comparison of Low-Level and High-Level Actions** (W2 for `Reviewer iozr`): We conducted an ablation study comparing high-level versus low-level control on the CL_L1 task, testing three strong closed-source models under both conditions. All models exhibited a significant drop in success rates when required to perform fine-grained navigation and control. This further suggests that using high-level actions to evaluate the model's capabilities is effective.

This summary provides an overview of the rebuttal stage for KidGym. We hope it proves helpful for the AC's review. We also want to express our gratitude for your time and effort in reviewing our paper.

Kind regards, KidGym Team

---

### Meta-Review · Area_Chair_iQNe · 2026-01-07

**Summary:**

This paper introduces KidGym, a 2D grid-based Gym-style benchmark for evaluating multimodal large language models across five cognitively inspired capabilities. The main strengths are the unified and interactive evaluation framework, the use of programmatic randomization to reduce data leakage, and the clear decomposition into capability dimensions with difficulty scaling. The main weaknesses are that the psychometric grounding remains conceptual rather than formally validated, some task–capability mappings rely on design intuition rather than empirical isolation, and several empirical findings reaffirm known limitations rather than introducing new failure modes. Overall, I recommend acceptance for this paper as it indeed introduces a useful evaluation benchmark, and the rebuttal substantially strengthened the benchmark’s rigor.

**Reviewer Concerns:**

Most concerns regarding experimental details were largely addressed through new ablations and experiments. While concerns about limited novelty (dWpT) remain only partially mitigated.

**Reviewer Scores:**

* Reviewer fF2z explicitly stated they increased their score after the rebuttal.
* Reviewer iozr indicated they would raise their evaluation if concerns were addressed. Given the rebuttal directly targets those issues, a score increase is likely.
* Reviewer dWpT and nziT will likely retain a similar score or slightly increase, with some concerns addressed through rebuttal.

---

### Decision · Program_Chairs · 2026-01-26

Accept (Poster)